# Understanding the Uncertainty of LLM Explanations: A Perspective Based on Reasoning Topology

**Longchao Da[†], Xiaoou Liu[†], Jiaxin Dai[†], Lu Cheng[‡], Yaqing Wang[§], Hua Wei[†*]**
[†]Arizona State University, [‡]University of Illinois Chicago, [§]Purdue University
{longchao,xiaoouli,jdai33,hua.wei}@asu.edu
{lucheng}@uic.edu, {wang5075}@purdue.edu

## Abstract

It is important to understand the uncertainty in large language models (LLMs) explanations because they reveal more information about the reasoning process and thus provide insights into the reliability of LLMs' answers regarding a question. In this work, we propose a novel framework that quantifies uncertainty in LLM explanations through reasoning topologies. By designing a structural elicitation strategy, we guide the LLM to frame the explanations on how it derives the answers into graph topologies. This strategy first queries knowledge-related sub-question and sub-answer pairs, and then guides the LLM to connect the pairs through a topological reasoning process. Based on the reasoning topologies, we revisit the Graph Edit Distance and provide a variant that can better quantify the LLM uncertainty from both semantic and reasoning structure dimensions. The topology structure further brings convenience to assess redundancy by extracting and comparing the valid reasoning path to the raw explanation. Extensive experiments show the effectiveness of the proposed framework, and interesting findings on reasoning patterns and efficiency are discussed.

## 1 Introduction

Deep learning models have long been criticized for their lack of trustworthiness due to the complex network structures and opaque decision-making processes (Li et al., 2022; Doshi-Velez & Kim, 2017; Samek et al., 2021). This has motivated researchers to investigate methods for understanding and quantifying the uncertainty associated with these models (Abdar et al., 2021; Loquercio et al., 2020; Maddox et al., 2019). Recently, Large Language Models (LLMs) have demonstrated significant advancements over traditional deep learning approaches across a variety of tasks (Zhao et al., 2023; Naveed et al., 2023). However, concerns about their reliability persist. LLMs often produce outputs that are difficult to verify, particularly in scenarios requiring complex reasoning (Shi et al., 2024). This introduces risks in critical applications, such as healthcare or legal contexts (Cascella et al., 2023; Jayakumar et al., 2023), where incorrect or unreliable reasoning can have severe consequences. Properly quantifying uncertainty in reasoning is crucial for ensuring safe and trustworthy reference.

Existing research on Uncertainty Quantification (UQ) for LLMs primarily focuses on analyzing semantic uncertainty (Kuhn et al., 2023; Lin et al., 2023; Qiu & Miikkulainen, 2024; Da et al., 2024), which involves examining patterns from the meaning of multiple responses generated for a given question. Although this can provide insights into output-level variability, it overlooks the structural characteristics of the reasoning process that give rise to such uncertainty. In other words, existing UQ methods do not model how uncertainty emerges and propagates through intermediate reasoning steps, which hinders practitioners' analyses, understanding, and potential remedy-based improvement. For instance, in Figure. 1 (right), when asked the same question, In part ① answer-based UQ can not reveal the vulnerable parts in the reasoning path, which are contained in the reasoning steps as in part ②.

---

*Corresponding Author

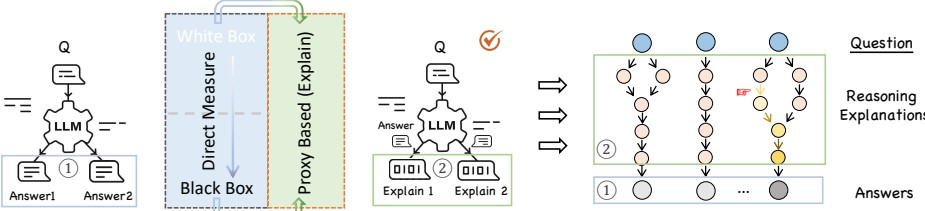

Figure 1: The comparison of Answer-based-UQ measures and Proxy (Explanations)-based UQ measures. The Explanations-based approach reveals more reasoning information.

By quantifying uncertainty at the reasoning dimension, we can effectively detect inconsistencies in intermediate steps, support human-in-the-loop validation in sensitive applications, and uncover vulnerabilities within the reasoning process itself. This underscores the importance of reasoning-aware uncertainty quantification. In this paper, we address this challenge by explicitly modeling reasoning processes as logical topologies. While prior work treats a single Chain-of-Thought (CoT) sequence as a monolithic response (Wei et al., 2022; Wang et al., 2023) and measures semantic consistency at the output level, such approaches fail to capture the structural complexity of real-world reasoning, which frequently involves hierarchical dependencies and parallel sub-tasks. To overcome these limitations, we introduce formal topology that represents reasoning as a logical graph, where nodes correspond to individual reasoning steps and edges encode their logical dependencies. This graph-based structure enables finer-grained, more interpretable uncertainty analysis.

Based on the structural representation, we introduce a graph-based measure for assessing uncertainty, by first encoding the node and edge descriptions into the semantic embeddings, and then performing a graph-edit-distance comparison, our framework captures the uncertainty from both semantic and topology aspects. Besides, we also propose a redundancy measure for the valid reasoning path, which helps understand the LLM's reasoning effectiveness. Extensive experiments on diverse datasets and LLMs demonstrate the utility of the proposed quantification methods. In summary, the contributions of this paper are:

- We identify limitations in existing answer-based UQ approaches for LLMs and propose a novel topology-elicitation framework that explicitly models the reasoning process from explanations as a logical topology.

- We introduce **Topo-UQ**, a topology-based uncertainty measure from both semantic meaning and reasoning structure to show granular thinking variance. We also propose a redundancy metric based on valid reasoning paths and raw explanations.

- We demonstrate the effectiveness of our framework through extensive experiments across multiple datasets and LLMs. Our results highlight its ability to identify inconsistencies in reasoning paths, improving trustworthiness and interpretability. Additionally, we discover three widely adopted reasoning patterns in LLMs and show the chance of improvement under the redundancy measure.

## 2 Related Work

In this section, we review the related work in the research domains of uncertainty quantification (UQ) for large language models (LLMs) and methods for explanation-based UQ, with a focus on reasoning processes.

### 2.1 UQ for LLM

**White-box Approaches** A significant body of research has focused on performing UQ for LLMs by inducing the models to output their uncertainty along with their responses Kadavath et al. (2022); Lin et al. (2022); Mielke et al. (2020); Tian et al. (2023). These methods often rely on token-level probabilities to train or fine-tune models for predicting uncertainty. While effective, these approaches require full access to the model's structure and weights,

which is impractical for black-box or commercial LLMs. For example, supervised methods such as those in Kadavath et al. (2022) estimate uncertainty using logits and ground truth labels but are computationally expensive and resource-intensive.

**Black-box Approaches**   Another line of work estimates uncertainty directly at the response level using semantic entropy Kuhn et al. (2023). While this method avoids token-level dependencies, it still relies on access to token probabilities, limiting its applicability in black-box settings. To address these limitations, researchers have proposed lightweight black-box methods that analyze response inconsistencies. For instance, Lin et al. (2023) uses graph Laplacian eigenvalues as an uncertainty indicator, while Chen & Mueller (2023) computes confidence scores from generated outputs to identify speculative or unreliable answers. However, these approaches primarily focus on semantic-level analysis and neglect the logical structure underlying reasoning processes. Moreover, methods like Lin et al. (2023) average entailment probabilities without considering directional information in reasoning paths.

Our work is agnostic to the white box or black box since it leverages the generated explanations as a proxy to measure the reasoning uncertainty as in Figure. 1. This enables a more nuanced and interpretable assessment of uncertainty in reasoning processes.

## 2.2   UQ for LLM Explanation

Explanation-based UQ focuses on assessing the reliability of natural language explanations (NLEs) generated by LLMs by either prompting models to express confidence in their explanations or analyzing consistency across multiple outputs under varying conditions Tanneru et al. (2024); Yadkori et al. (2024). While these methods provide insights into explanation robustness, they treat explanations to a question as unstructured text representation, which lacks structural information and fails to capture inconsistencies or leaps in logic. In contrast, our work explicitly leverages well-structured reasoning topologies to enhance the UQ process for explanations. This structured representation enables us to assess explanation uncertainties at a finer granularity within complex reasoning paths.

## 3   Preliminaries

This section provides the foundations for our study, including the definition of uncertainty in LLMs and its quantification in natural language explanations (NLE) fashion.

### 3.1   Uncertainty Quantification of LLMs

Uncertainty, as an inherent attribute of LLMs, can arise from various aspects: training corpus coverage (Huang et al., 2023), prompt ambiguity (Abbasi Yadkori et al., 2024), the probabilistic nature of language generation (Lin et al., 2023), and many more. Effectively modeling and interpreting uncertainty are essential for reliable and trustworthy references; thus, researchers quantify the uncertainty for better decision-making.

Different from UQ methods in classical machine learning (Lakshminarayanan et al., 2017; Gal & Ghahramani, 2016; Hernández-Lobato & Adams, 2015; Abdar et al., 2021), UQ in LLM faces unique challenges such as discrete token-based generation, the large combinatorial output space, and expression diversities of the same concept, thus, most LLM-UQ methods focus on analyzing uncertainty from model-generated responses as shown in ① in Figure. 1. The problem could be defined as follows:

**Problem 1** (Uncertainty Quantification for LLMs). *The LLM 'M' is provided with an input query $x^q$, the goal of the uncertainty function $U_x$ is to map the generated outputs to a scalar score that determines the uncertainty of the M, i.e., $U_x = \mathcal{U}\left(\{M(x_i^q)\}_{i=1}^n\right)$*

Here, $\{M(x_i^q)\}_{i=1}^n$ denotes a set of $n$ responses[1] generated by the model $M$, and $\mathcal{U}$ aggregates uncertainty across multiple responses. $U_x$ is calculated only on $x$ and is a property of the predicted distribution, which is estimated by $\mathcal{U}$. For black-box methods that analyze response variability or semantic consistency (Lin et al., 2023), multiple outputs ($n > 1$) are typically required. In contrast, white-box methods that rely on internal model parameters, such as logits (Kadavath et al., 2022), may only require a single output ($n = 1$). However, both current answer-based black-box Lin et al. (2023); Chen et al. (2025); Su et al. (2024) and white-box methods Margatina et al. (2023); Duan et al. (2024); Kadavath et al. (2022) share limitations in the quantification of reasoning procedures in that they focus on semantic variability without capturing deeper uncertainties in reasoning steps.

## 3.2 Uncertainty of LLM Explanations

To address these issues, researchers try to understand uncertainties in reasoning processes through the *natural language explanations (NLEs)* as a proxy, introduced by Camburu et al. (2018); Tanneru et al. (2024). As shown in the green part (②) of Figure. 1, an NLE is a textual reasoning sequence generated by a language model $M$, to justify or explain the derivation of answer $a$ for a given input question $x^q$. Following the existing work, it is defined as:

**Definition 1** (Natural Language Explanation). *Given a model $M$ and an input query $x^q$, an NLE can be represented as:*

$$M(x^q + x^e) = a + a^e \tag{1}$$

*where $x^e$ is an explanation-specific prompt, $a$ is the model's answer to the query $x^q$, and $a^e$ is the generated explanation accompanying the answer. The explanation $a^e$ contains a sequence of reasoning steps, represented as $a^e = \{s_1, s_2, ..., s_m\}$, which capture the reasoning process or justification for $a$.*

To quantify uncertainty through explanations, we extend Problem 1 to consider $n$ explanations generated for the same query $x^q$. Each explanation $a_i^e$ ($i \in \{1, 2, \ldots, n\}$) corresponds to a set of reasoning steps, assuming $m$ steps, then we denote them as $a_i^e = \{s_{i,1}, s_{i,2}, ..., s_{i,m}\}$. The overall uncertainty across all $n$ explanations is captured by aggregating reasoning-level uncertainties for each explanation. This can be formally defined as:

**Problem 2** (Uncertainty Quantification for LLM Explanations). *Given an input query $x^q$ and an explanation-specific prompt $x^e$, the model $M$ generates a set of answers $a_i$ to the query $x^q$, along with accompanying explanations $a_i^e$. The uncertainty for the query $x^q$ is then defined as:*

$$U_{x^q} = \mathcal{U}\left(\bigcup_{i=1}^n a_i^e\right) = \mathcal{U}\left(\bigcup_{i=1}^n \{s_{i,1}, s_{i,2}, ..., s_{i,m}\}\right) \tag{2}$$

*where $U_{x^q}$ represents the overall uncertainty for the query $x^q$, and $\mathcal{U}$ aggregates uncertainties across all reasoning steps from all $n$ explanations.*

Unlike prior methods that focus on answer-based output variability (Tanneru et al., 2024), in this paper, we aim to capture nuanced uncertainties from the reasoning process, which brings advantages in analyzing the reasoning patterns and logical errors. In the next section, we will introduce **Topo-UQ**, including how to model reasoning structures within explanations, and how this graph-based UQ method works with the extracted structures.

## 4 Method

In this section, we propose a reasoning **Topo**logy-based method for **U**ncertainty **Q**uantification of LLMs, named **Topo-UQ**. To capture the complexity of reasoning paths, the proposed framework consists of two modules: (1) *Reasoning Topology Elicitation*, which constructs a structured reasoning graph from LLM-generated explanations, and (2) *Topology-based UQ*, which leverages the constructed graphs for uncertainty quantification using a variant of graph edit distance. We also present a byproduct - a metric of reasoning path redundancy inspired by the analysis of reasonable reasoning paths and original explanations.

---

[1]Answers and responses are used interchangeably in this paper.

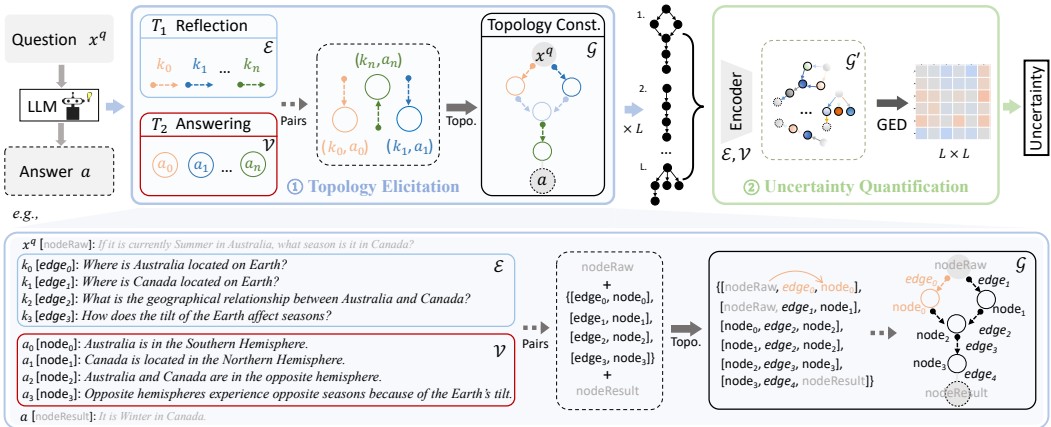

Figure 2: An overview of **Topo-UQ**. It depends on two modules, ① `Topology Elicitation` and ② `Uncertainty Quantification`. Start from ①, given an input question $x^q$, the LLM not only generates the answer $a$ but is also elicited to provide reasoning explanations. It first reflects the necessary knowledge points ($k_n$) as sub-questions and then answers them as ($a_n$), $T_1$ and $T_2$ are the prompt templates. These sub-question and answer sets form the edge and node sets $\mathcal{E}, \mathcal{V}$, creating matched pairs. By inserting a node into an (edge-node) pair, the LLM constructs the reasoning topology $\mathcal{G}$. When we have $L$ explanations, their topologies are used to estimate **Uncertainty** through module ②: we encode the content from each node and edge into embedding space to capture semantic meanings, and apply a variant of Graph Edit Distance to measure structural variance, then the final uncertainty combines both semantic and structural differences. An example is shown at the bottom.

## 4.1 Reasoning Topology Elicitation

The objective of this step is to derive a structured reasoning topology that captures the complexity of reasoning paths generated by large language models (LLMs). Existing approaches by Tanneru et al. (2024) represent reasoning explanation as linear sequences of steps from Chain-of-Thought (CoT) (Wei et al., 2023). While linear text sequences provide basic interpretability, they fail to capture complex logical transitions in tasks like comparative reasoning or multi-faceted conclusive answering. This lack of structural richness limits their ability to analyze and quantify uncertainty in LLM-generated reasoning processes.

To address these limitations, we propose to elicit reasoning topologies from a question $x_q$ to an answer $a$ as a directed graph $\mathcal{G} = (\mathcal{V}, \mathcal{E})$, where $\mathcal{V}$ is the set of nodes corresponding to knowledge points or intermediate steps, and $\mathcal{E}$ is the set of edges capturing logical dependencies between them. The $\mathcal{G}$ includes two advantages: **First**, it can represent diverse structures, not limited to the linear sequence CoT. **Second**, the construction is done by the LLM itself, thus it can flexibly reflect the thinking process for each answer. For example in Figure. 2, given the query $x^q$: " *If it is currently Summer in Australia, what season is it in Canada?* ", the knowledge points might include sub-questions such as $k_0$: 'Where is Australia located on Earth?', $k_1$: 'Where is Canada located on Earth?', $k_2$: 'What is the geographical relationship between Australia and Canada?' and $k_3$: 'How does the tilt of the Earth affect seasons?'. The corresponding answers $A = \{a_0, a_1, a_2, a_3\}$ demonstrate thoughts of LLM for sub-questions. These knowledge-answer pairs are then connected based on their logical dependencies to form the reasoning topology graph $\mathcal{G}$. The connection becomes a step from $a_i$ to $a_j$: $S = \{[a_i, a_j, k_{ij}] \mid a_i, a_j \in \mathcal{V}, k_{ij} \in \mathcal{E}\}$

Specifically, the construction of $\mathcal{G}$ consists of three steps: (1) *Knowledge Point Reflection*, where the LLM identifies sub-questions of knowledge points $k_n$ required to address the query as elicited by the template $T_1$; (2) *Self-Answering Module*, based on the sub-questions among $k_n$, the LLM generates answers for them, denoted as $a_n$, in this process, template $T_2$ carries the question variable which can be changed according to different sub-questions; up to

this point, there is an intermediate set containing pairs[2] $(k_n, a_n)$, which will be used for (3) *Reasoning Topology Construction*. In this part, the knowledge-answer pairs are connected into a directed graph that reflects the overall reasoning process. It mainly requires an insertion from the current node-edge pair to a prior node, which is identified by LLM itself. The detailed explanations of every module and prompt templates are introduced in the Appendix. B.

## 4.2 Topology-based Uncertainty Quantification

With the `Topology Elicitation` module built, we intend to understand the uncertainty from the elicited structural explanations. Given a query $x^q$, the model $M$ will be asked $L$ times, from which reasoning topologies $\{\mathcal{G}_i^q\}_{i=1}^L$ will be extracted. We propose a UQ method: **Reason**ing **G**raph **E**dit **D**istance, namely **Reason-GED**, to evaluate the consistencies of $\{\mathcal{G}_i^q\}_{i=1}^L$ from the semantics embeddings of node, edge sentences, and structural topologies.

If we apply the traditional graph distances (Bai et al., 2020; Huang et al., 2024; Guzman-Pando et al., 2024) that solely quantify the differences in structures, we will lose the node and edge meaning implied in the structure. To understand the structural variance without loss of semantic meaning, we first encode the node and edge content into semantic embeddings.

**Step1: Semantic Embedding.** To measure semantic meanings of reasoning steps, for each graph $\mathcal{G} = (\mathcal{V}, \mathcal{E}) \in \{\mathcal{G}_i^q\}_{i=1}^L$ we employ an embedding function $\mathcal{L}$ to encode the content of nodes and edges in graph. Since the content of each node $v \in \mathcal{V}$ and edge $e \in \mathcal{E}$ is textual description, we can derive embeddings:

$$\mathbf{h}^v = \mathcal{L}(v), \quad \mathbf{h}^e = \mathcal{L}(e), \quad \forall v \in \mathcal{V}, e \in \mathcal{E} \tag{3}$$

This paper adopts BERT as the embedding function $\mathcal{L}$, but other embeddings could be preferred for different domain contexts, e.g, (Rasmy et al., 2021) for medical text. The encoded graph $\mathcal{G}'$ is shown in the gradient color of ② in Figure. 2. This obtains the semantics of nodes and edges while preserving the logical structure of the reasoning process.

**Step2: Reasoning Distance from Topology and Semantic.** In our setting, we have $L$ reasoning structures $\{\mathcal{G}_i^q\}_{i=1}^L$ generated. To measure the pairwise distance of two reasoning processes $\mathcal{G}_1$ and $\mathcal{G}_2$, inspired by the concept of graph edit distance (Gao et al., 2010), we use the minimum transformations (substitution, deletion, insertion) required to align the two graphs as their pairwise distance.

*A. Substitution Costs:* For two corresponding (edge, node) pairs in different topologies $\mathcal{G}_1 = (\mathcal{V}_1, \mathcal{E}_1)$ and $\mathcal{G}_2 = (\mathcal{V}_2, \mathcal{E}_2)$ we define the semantic substitution cost based on $(v_i, e_i) \in \mathcal{V}_1, \mathcal{E}_1$ and $(v_j, e_j) \in \mathcal{V}_2, \mathcal{E}_2$:

$$c\{(e_i, v_i), (e_j, v_j)\} = \begin{cases} 1 - \cos(\mathcal{L}(e_i), \mathcal{L}(e_j)) = 1 - \cos(\mathbf{h}_i^e, \mathbf{h}_j^e), & \text{edge substitute} \\ 1 - \cos(\mathcal{L}(v_i), \mathcal{L}(v_j)) = 1 - \cos(\mathbf{h}_i^v, \mathbf{h}_j^v), & \text{node substitute} \end{cases} \tag{4}$$

where cosine similarity measures the semantic alignment of two reasoning edges and nodes. If $c\{\cdot, \cdot\}$ is high, this indicates: either given a similar edge (sub-question), the nodes (sub-response) are different - there might exist an incorrect sub-answer, or difference starts with edges, which follow up with different responses - there is a different reasoning path or jumping step. The examples for these cases are shown in Appendix C.1.

*B. Deletion/Insertion Costs:* In order to decide which node/edge to delete or insert to align two reasoning topologies. We define the deletion and insertion costs. For a node $v_i \in \mathcal{V}_1$ in graph $\mathcal{G}_1 = (\mathcal{V}_1, \mathcal{E}_1)$ relative to target graph $\mathcal{G}_2 = (\mathcal{V}_2, \mathcal{E}_2)$:

$$c_{\text{del.}}(v_i) = \frac{1}{2} \Big\{ \underbrace{\max_{v_j \in \mathcal{V}_2} \cos\big(\mathcal{L}(v_i), \mathcal{L}(v_j)\big)}_{\text{Cross-graph Matching Term}} + \underbrace{1 - \frac{1}{|\mathcal{V}_1| - 1} \sum_{\substack{v_k \in \mathcal{V}_1 \\ v_k \neq v_i}} \cos\big(\mathcal{L}(v_i), \mathcal{L}(v_k)\big)}_{\text{Internal Uniqueness Term}} \Big\}. \tag{5}$$

---

[2] $(k_n, a_n)$ = (edge, node) = $(e, v)$

The Eq. 5 jointly considers the cross-graph matching cost and the internal uniqueness cost when deleting node $v_i$. In the cross-graph matching part, if $v_i$ has a very similar counterpart $v_j$ in $\mathcal{G}_2$, then deleting $v_i$ would lose important alignment information; thus, a high similarity here leads to a higher deletion cost. In the uniqueness part, if $v_i$ is very similar (i.e., redundant) to many other nodes, the average similarity will be high, making this term low. In other words, deleting a redundant node is less costly. Otherwise, if $v_i$ is unique, deletion is costly. Since the insert. is a reverse action of delete., they share the same equation. The above defines the cost function for a node, the same applies to edge $e_i$, complete Eq. 19 is in Appendix C.2.

**Step3: Graph Distance for Reasoning Uncertainty.** Based on the above two steps, we can derive the overall graph edit cost in joint consideration of semantic meaning and topology variance as: $\text{GED}(\mathcal{G}_1, \mathcal{G}_2) = C_{\text{sub.}}(\mathcal{P}) + C_{\text{del.}}(\mathcal{P})$, the $\mathcal{P}$ represents the optimal matchings for sub-questions (edges $\mathcal{P}_e$) and sub-responses (nodes $\mathcal{P}_v$), using the Hungarian algorithm (Mills-Tettey et al., 2007), details are explained in the Appendix. H.1. We can calculate the minimal total cost of transformations by finding:

$$\text{GED}_m(\mathcal{G}_1, \mathcal{G}_2) = \min_{\mathcal{P}} \text{GED}(\mathcal{G}_1, \mathcal{G}_2) \tag{6}$$

Given this, a higher GED implies a higher difference in the reasoning phase by considering both embedding and structures. We use this computed reasoning distance and construct a matrix across $L$ reasoning structures $\{\mathcal{G}_1, \mathcal{G}_2, \ldots, \mathcal{G}_l\}$, we compute pairwise distances using: $d_{ij} = \text{GED}_m(\mathcal{G}_i, \mathcal{G}_j)$, which then forms the overall distance matrix between $L$ reasoning topologies as shown on the right of Figure. 2: $D^{\mathcal{G}_n} = [d_{ij}] = [\text{GED}_m(\mathcal{G}_i, \mathcal{G}_j)]_{L \times L}$. Now, we can calculate the **Topo-UQ** by the variance of the distances in $D^{\mathcal{G}_n}$, which reflects instability of the model's reasoning behavior. Combining Eq. 2, we have the uncertainty score over a query $x^q$ as: $\mathcal{U}_{\text{struct}}(x^q) = \text{Var}(D^{\mathcal{G}_n})$ A higher variance indicates greater inconsistency in the model's reasoning, suggesting that the LLM generates more different structures across multiple responses to the same query.

### 4.3 LLM Reasoning Redundancy Measure

It is known that the LLM's reasoning efficiency varies based on the problem type and model weights (Plaat et al., 2024), and the reasoning topology provides a good reference to understand the efficiency by analyzing the valid reasoning path. We find that LLMs do not necessarily rely on all of the nodes from their reasoning topology for the final conclusion, which means some of the sub-steps do not contribute to solving a problem, and this causes the decreased efficiency. Here, we propose a measurement named 'Reasoning Redundancy'. Reflect the reasoning steps: $S = \{[v_i, v_j, e_{ij}] \mid v_i, v_j \in \mathcal{V}, e_{ij} \in \mathcal{E}\}$, we aim to measure the redundancy based on the valid path constructed by the steps.

**Definition 2** (Redundant Node). *A node $v_k \in \mathcal{V}$ is redundant if it does not contribute to the reasoning path from nodeRaw to nodeResult. Formally, a redundant node satisfies:*

$$v_k \notin \bigcup_{[v_i, v_j, e_{ij}] \in \mathcal{P}_{valid}} \{v_i, v_j\} \tag{7}$$

*where $\mathcal{P}_{valid}$ represents the set of all valid paths contributing to the final conclusion, nodeRaw and nodeResult are the initial reasoning question and result, respectively, as in Figure. 2.*

Given the definition, it is feasible to detect the redundancy following the criteria in Appendix H.2, we perform the searching for $\mathcal{P}_{valid}$ valid paths and 'Redundancy Rate' using traversal algorithm (DFS), then we have the redundancy rate of the reasoning process for $a_i^e$:

$$r_{\text{redun.}}(a_i^e) = \frac{|\mathcal{V}_{\text{redundant}}|}{|\mathcal{V}|} \tag{8}$$

where the $|\mathcal{V}|$ is the total number of nodes in the reasoning topology, and $|\mathcal{V}_{\text{redundant}}|$ is the number of redundant nodes (isolated from a valid reasoning path).

## 5  Experimental Study

In this section, we conducted extensive experiments covering 5 LLMs: GPT4o-mini, DeepSeek-R1 (distilled on llama3-70b), Phi4, Llama3-8b, Llama3-70b (v3.3), 3 classic datasets, and 5 UQ methods. We intend to answer the following research questions:

- RQ1: Can **Topo-UQ** reveal the actual uncertainty in LLM's reasoning?

- RQ2: How do LLMs perform in the proposed redundancy measure?

- RQ3: From the proposed topology reasoning elicitation, do LLMs share certain patterns commonly?

### 5.1  Experiment Settings

**Dataset:** In this study, to align with the research community (Mirzadeh et al., 2024; Toshniwal et al., 2024; Zhang et al., 2024), we utilized widely adopted datasets, including GSM8k (Cobbe et al., 2021) and BoolQ (Clark et al., 2019), which require complex reasoning rather than simple question-answering. These benchmarks assess LLM's ability to perform multi-step inference and logical reasoning. Besides, we also develop a new dataset, GeoQA, especially for condition-based reasoning tasks. We will explain the details of the GeoQA dataset in Appendix E and provide a brief introduction of all datasets in Appendix I.1

**Baselines:** To the best of our knowledge, few works focus on the uncertainty quantification of natural language explanations (**NLE**), thus, we included all possible baselines: (1) Chain-of-Thought Agreement (CoTA) (Tanneru et al., 2024), (2) Embedding distance-based UQ for NLE (Embed-UQ), (3) Entailment probability-based (Entail-UQ), and (4) NLI-logit based UQ, as our extra baselines to understand their interpretability of LLM explanations. We provide a brief introduction here, and for the detailed explanation of baselines, please refer to Appendix. G.

**Evaluation and Metrics:** To evaluate the performance of uncertainty quantification methods in LLM explanation tasks, we follow the standard practice that compares uncertainty results with actual faithfulness (Tanneru et al., 2024). The ground truth faithfulness score reveals how much the model relies on its complete reasoning process, which is calculated through a strategy named 'Early Answering' as proposed by (Lanham et al., 2023), We provide details on how the faithfulness score is derived in Appendix I. **Ideally, a UQ method is good in NLE if, for a higher faithful set, it generates lower uncertainty, and vice versa** (Tanneru et al., 2024). Hence, we employ three robust statistical metrics to quantify the correlation between the derived uncertainty and faithfulness. Including Pearson Correlation Coefficient (PCC), which is used to measure the linear correlation between two variables. Given the relatively small amount of each bootstrap sample, we employ two extra metrics, Spearman Rank Correlation (SRC) and Kendall Rank Correlation (KR); the calculation of metrics is also in the Appendix J.

For fair evaluation and to avoid bias, we conduct a bootstrap for a given dataset $D_{test}$ and measure the correlation between uncertainty and faithfulness score at each sub-set $D'$ level. The sub-set is cut as 20 questions with 10 responses for each question = 200 $a^e$, and bootstrapping is conducted 1000 times on each dataset.

### 5.2  Quantitative Evaluation (RQ1)

In order to understand if Topo-UQ can reveal the uncertainty in LLMs' reasoning, we perform experiments on GSM8K, BoolQ, and GeoQA datasets. Due to the page limit, GSM8K and BoolQ are shown in the Table. 1, GeoQA is shown in Appendix, Figure. 4. Our method reveals a stronger negative correlation between the derived UQ results and the groundtruth faithfulness across different statistic metrics, the results on GPT4o-mini, Llama3-70b, and DeepSeek-R1 are more convincing because they have a more stable performance on the Topology elicitation task as in Figure. 6, which, is a key step for the proposed UQ method, the performance on Phi4 and Llama3-8b not promising as ranked in the last two positions

| Methods | GPT4o-mini | | | DeepSeek-R1 | | | Llama-3.3-70B | | | Llama3-8b | | | Phi4 | | |
|---|---|---|---|---|---|---|---|---|---|---|---|---|---|---|---|
| | PCC(↓) | SRC(↓) | KR(↓) | PCC(↓) | SRC(↓) | KR(↓) | PCC(↓) | SRC(↓) | KR(↓) | PCC(↓) | SRC(↓) | KR(↓) | PCC(↓) | SRC(↓) | KR(↓) |
| **Dataset: BoolQ** | | | | | | | | | | | | | | | |
| CoTA | 0.03 | 0.05 | 0.03 | -0.22 | -0.20 | -0.13 | -0.00 | 0.05 | 0.03 | -0.07 | -0.06 | -0.04 | 0.08 | 0.07 | 0.05 |
| Embed-UQ | 0.16 | 0.18 | 0.12 | 0.56 | 0.56 | 0.39 | 0.09 | 0.12 | 0.08 | 0.04 | 0.06 | 0.04 | 0.02 | 0.02 | 0.01 |
| Entail-UQ | **-0.12** | **-0.13** | **-0.09** | 0.48 | 0.46 | 0.32 | -0.04 | -0.08 | -0.05 | -0.08 | -0.08 | -0.05 | **-0.09** | **-0.09** | **-0.06** |
| NLI-logit | 0.19 | 0.18 | 0.12 | 0.56 | 0.56 | 0.39 | 0.07 | 0.10 | 0.07 | 0.03 | 0.03 | 0.02 | -0.01 | -0.01 | -0.01 |
| Ours | -0.03 | -0.05 | -0.03 | **-0.29** | **-0.26** | **-0.17** | **-0.24** | **-0.20** | **-0.17** | **-0.24** | **-0.23** | **-0.15** | 0.01 | 0.01 | 0.01 |
| **Dataset: GSM8K** | | | | | | | | | | | | | | | |
| CoTA | -0.12 | -0.10 | -0.07 | **-0.40** | **-0.39** | **-0.26** | -0.13 | -0.13 | -0.09 | -0.04 | -0.03 | -0.02 | **-0.14** | **-0.13** | **-0.09** |
| Embed-UQ | -0.12 | -0.10 | -0.07 | 0.09 | 0.10 | 0.07 | 0.15 | 0.14 | 0.10 | 0.23 | 0.23 | 0.15 | 0.28 | 0.28 | 0.19 |
| Entail-UQ | 0.14 | 0.14 | 0.09 | 0.68 | 0.66 | 0.47 | 0.15 | 0.13 | 0.09 | -0.08 | -0.07 | -0.05 | 0.07 | 0.07 | 0.05 |
| NLI-logit | 0.00 | 0.01 | 0.01 | 0.10 | 0.11 | 0.07 | 0.13 | 0.12 | 0.08 | 0.21 | 0.19 | 0.13 | 0.29 | 0.29 | 0.20 |
| Ours | **-0.35** | **-0.34** | **-0.23** | -0.22 | -0.20 | -0.14 | **-0.43** | **-0.41** | **-0.28** | **-0.14** | **-0.13** | **-0.08** | 0.12 | 0.10 | 0.06 |

Table 1: Comparison of our methods with different baselines on various datasets and large language models. The results show that our results consistently outperform the baseline methods. The value reflects the correlation between the uncertainty and the ground truth faithfulness; the more faithful an LLM performs, the less uncertainty it should have, so by the UQ methods, we would expect a negative correlation for a well-performed UQ method.

in Appendix Figure. 6. This RQ result shows our method is effective in revealing the LLM's real faithfulness, yet it is more suitable for LLMs with good instruction-following abilities.

## 5.3 Redundancy Measure of LLMs (RQ2)

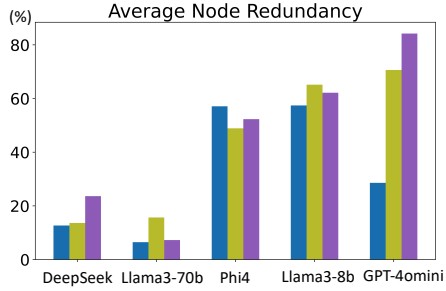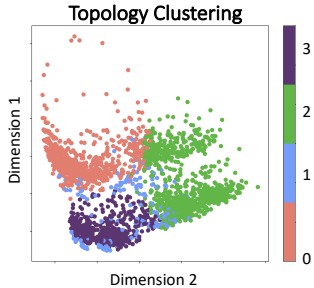

Figure 3: The left side shows the quantification of the LLM's reasoning redundancy on three datasets: GeoQA (Blue), GSM8K (Yellow), BoolQ (Purple); The right side shows the clustering result of the reasoning topologies.

Benefitting from the topology structure, we are able to extract the reasoning path $\mathcal{P}_{\text{valid}}$ that successfully connects from nodeRaw to nodeResult. Then, we effectively search the node that does not contribute to the final answer, which serves as a sign of redundancy in the reasoning process. Following the Eq. 8, we analyze the redundancy rate for each of the five LLMs on both the node and edge redundancy. Since the node and edge always show up together in our setting, their redundancy results are very similar; we just present redundancy on the 'node' level for conciseness in the left part of Figure. 3. We find that, surprisingly, the GPT-4o-mini shows a significantly high redundancy rate in both the nodes and edges, which might reveal that the high accuracy of the model comes from a broader searching space (or generating length) when conducting reasoning and proposing solutions. However, this also reflects that there would be great potential to optimize the reasoning process for the GPT-wise models. Comparatively, the DeepSeek (distilled llama version) is relatively low, which might indicate the model's training was conducted with a special design to encourage the 'valid' reasoning, which eventually contributes to the final result. We show more results in the appendix for reference.

## 5.4 Findings on the Reasoning Patterns from LLMs (RQ3)

Based on the topology representations, we conducted further analysis in structural patterns, we use Graph2Vec (Narayanan et al., 2017) to embed the graph structure information, such as node connectivity, and perform K-means clustering on 4 classes and obtained silhouette score of 0.38, as shown in right part of Figure. 3, we found three major reasoning patterns of the LLMs (in color - red, green, purple on the right side of Figure. 3), they correspond with:

① Completion-based: the model has seen the data in the training phase, and completes it as a 'cloze question'. For example, given the question that provides a feature of climate type, and asks for the answer as shown in Figure. 4, then in the sub-question step, the LLM directly asks 'Does object A have such a feature?', it shows a one-step, direct answer which possibly comes from the training memorization. ② Forward reasoning (actual reasoning): The model is actually performing reasoning based on the analysis procedure, like human beings, given a question, it first recalls relevant concepts and knowledge that contain the necessary information. Then, through a step-by-step narrowing-down process, it refines its understanding, gradually converging on the most precise answer. ③ Combination of Memory-based answering, forward reasoning (① + ②) + verify: The model uses two methods and double-checks to verify the answer. Three modes are shown in Figure. 4. We hope this could serve as an inspiration for the potential further study in answer topology.

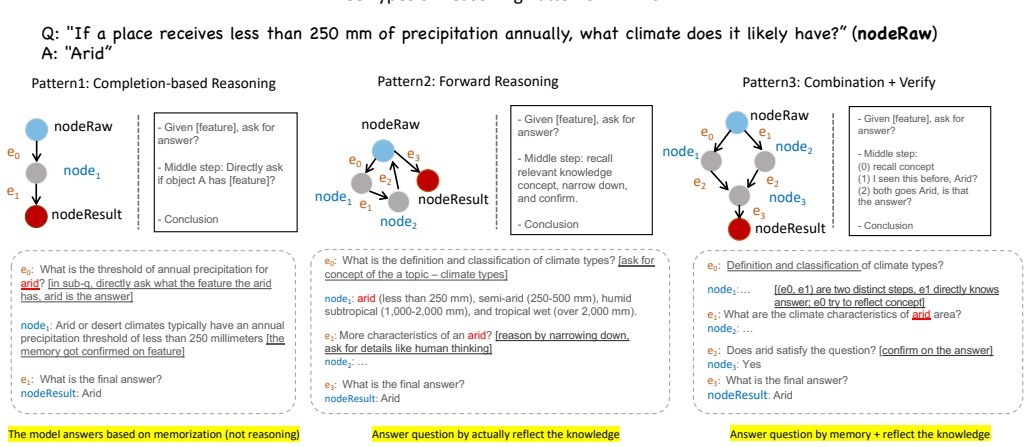

Figure 4: The examples of three types of reasoning patterns. As shown in the image, the first type directly reflects the correct answer in its first reasoning step, and not actually performing the reasoning, the second type is like humans, try to recall the knowledge points, and narrow down the range, and find the possible answer in a thought chain, while the third one is a combination of two, it has the steps to ask for characteristics, but it also try to reflect if it has seen this question before at the edge 1, which is a direct attempt to use answer to match the question, the last step is to confirm if the memory of answer matches the requirement in the question.

## 6 Conclusion

In this paper, we introduce a novel framework **Topo-UQ** to quantify the uncertainty of LLMs' reasoning through the explanations. By designing a reasoning topology elicitation module, we tackle the challenge of shaping the structural reasoning process of LLMs. Then, based on the constructed topologies, the paper provides a 'graph edit distance' based uncertainty measure named **Reason-GED**. It considers both the semantic meanings of the reasoning steps and the structural topology variance, by a set of well-designed operation costs: substitution and deletion (insertion), we can effectively quantify the reasoning uncertainty. The empirical studies show that this method provides better performance in revealing the true faithfulness of the natural language models. Besides, this paper also introduces a redundancy-based method to quantify the conciseness of the LLMs' reasoning process, which could potentially serve as a guide for improving reasoning efficiency.

**Limitations**: Although the **Topo-UQ** shows an interesting direction (topology structure) to quantify the LLM reasoning uncertainty, it requires the LLMs to come with good instruction-following ability. Thus, this method is more applicable to larger models with long-context reasoning capabilities. Besides, the current method is validated by the faithfulness, but it does not concern the answer's correctness from the reasoning path; analyzing the uncertainty relevant to the correctness would also be beneficial for real-world use-cases.

## Ethics Statement

This paper focuses on the uncertainty quantification of the LLM explanations. The data adopted are the widely acknowledged public datasets, and contain no ethics sensitive information, which has no ethics concerns.

## Reproducibility Statement

This project will release the codes for the topology elicitation that is used for uncertainty quantification, and we conducted pre-processing for the three datasets, which will also be updated to the repository: https://github.com/LongchaoDa/LLM-Topology.

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

# A   Notation Table

In this section, we provide a comprehensive notation table as shown in the Table. 2.

| Notation | Explanation |
| --- | --- |
| $U_x$ | Uncertainty function |
| $M$ | Large language model |
| $x$ | Input(query or prompt) |
| $\mathcal{U}$ | Aggregation |
| $x^q$ | Input prompt |
| $x^e$ | Explanation-specific prompt |
| $a$ | LLMs answer |
| $a^e$ | Generated explanation accompanying the answer $a$ |
| $s_i$ | Reasoning step |
| $\mathcal{G} = (\mathcal{V}, \mathcal{E})$ | Directed reasoning graph |
| $\mathcal{V}$ | Set of nodes |
| $|\mathcal{V}|$ | Number of nodes |
| $\mathcal{E}$ | Set of edges |
| $|\mathcal{E}|$ | Number of edges |
| $e_{ij}$ | Edge from node $i$ to node $j$ |
| $K^q$ | Knowledge points |
| $T$ | Prompt template |
| $\mathcal{D}_m$ | Set of knowledge-answer pairs for reasoning topology |
| $(k_i, a_i)$ | Knowledge-answer pairs |
| $e^F$ | Sampling $F$ amount of edges |
| $\mathcal{F}$ | Demonstration set |
| $\mathcal{L}(\cdot)$ | Embedding function |
| $\mathbf{h}$ | Contextual embedding |
| $cos(\cdot)$ | Cosine similarity |
| $\mathcal{P}$ | Optimal matching for sub-questions |
| $d_{ij}$ | Graph edit distance between $\mathcal{G}_i$ and $\mathcal{G}_i$ |
| $r_{redun.}$ | Redundancy rate |

Table 2: The notations and explanations in this paper.

# B   Reasoning Topology Costruction Process in Detail

## B.1   Knowledge Point Reflection Module

The first module, the knowledge point reflection module, involves eliciting sufficient information that can be used to support the conclusion drawing toward the input $x^q$. The input to this module is the input query $x^q$ along with the prompt template $T_1$ to encourage the LLM to reflect 'What knowledge basis (or sub-questions) it should know to draw a final conclusion?'. And the output of this module is the set of knowledge points $K^q$ extracted as a series of sub-questions, i.e., $K^q = \{k_1, k_2, \ldots, k_n\}$. Specifically, we design a prompt template $T_1$ to guide the model in reflecting on the sub-questions or knowledge points required for solving $x^q$:

> **Template $T_1$:** Given a question $\{x^q\}$, reflect and generate the knowledge points as sub-questions necessary to solve this query. Ensure that the output is both sufficient and concise.

The model generates a set of knowledge points $K^q = M(x^q, T_1) = \{k_1, k_2, \ldots, k_n\}$, where each $k_i$ corresponds to a specific sub-question or piece of information identified as necessary to address the query $x^q$ under the guidance of prompt $T_1$. To ensure traceability, we assign unique identifiers to each knowledge point using a tagging function $f(\cdot)$:

$$K^q_{tag} = \{\text{id}_1 : k_1, \text{id}_2 : k_2, \ldots, \text{id}_n : k_n\}. \tag{9}$$

In the later sections of this paper, we assume the $k^q$ always carries its identifier while performing computing ($K^q \Leftrightarrow K^q_{tag}$).

## B.2 Self-answering Module

To provide answers for the knowledge points $K^q = \{k_1, k_2, \ldots, k_n\}$ elicited in the previous module, we design a self-answering module to generate precise answers. For each sub-question $k_i \in K^q$, the model $M$ generates an answer $a_i$ using the following prompt $T_2$, ensuring coherence and sufficiency in addressing each of the knowledge points:

> **Template** $T_2$: Given a sub-question $\{k_i\}$, provide a precise answer that directly addresses the query without further discussion.

The model generates answers $A = \{a_1, a_2, \ldots, a_n\}$, where each answer $a_i = M(k_i, T_2)$. So we have:

$$A = \{a_1, a_2, \ldots, a_n\} = \{M\{k_1, T_2\}, M\{k_2, T_2\}, \ldots, M\{k_n, T_2\}\} \tag{10}$$

This formulation explicitly links each sub-question $k_i$ to its corresponding answer $a_i$, ensuring clarity in the relationship between the elicited knowledge points and their responses. The resulting set of knowledge-answer pairs forms the basis for constructing the reasoning topology:

$$\mathcal{D}_m = \{(k_1, a_1), (k_2, a_2), \ldots, (k_n, a_n)\} \tag{11}$$

## B.3 Reasoning Topology Construction Module

To construct the reasoning topology graph $\mathcal{G}^q = (\mathcal{V}, \mathcal{E})$, a critical step would be to connect the $(k, a)$ pairs in a structured format based on their logical dependencies. Since we are quantifying the uncertainty of LLM explanations, this connection should be determined by the model itself to explain. Therefore, in this module, we leverage the few-shot learning ability of LLMs and guide them in connecting the basis $(k, a)$ pairs following their reasoning procedure. By sampling $F$ amount of $e^F$ as few-shot examples from a demonstration set $\mathcal{F}$ and feeding them to the model, the LLM learns to depict the reasoning path in a structured way for this task[3]: $\hat{\mathcal{D}}_m = M(\mathcal{D}_m, e^F)$, the transformation from $\mathcal{D}_m$ to $\hat{\mathcal{D}}_m$ follows:

$$\hat{\mathcal{D}}_m = M(\mathcal{D}_m, e^F) = \{(a^{p_1}, k_1, a_1), (a^{p_2}, k_2, a_2), \ldots, (a^{p_n}, k_n, a_n)\} \tag{12}$$

where each $a^{p_i}$ is an answer node that connects to the corresponding knowledge-answer pair $(k_i, a_i)$.

To ensure that the reasoning path forms a structured yet flexible topology that adapts to the complexity of real-world cases, the specific ordering of $p_i$ is not predetermined and depends on the actual reasoning structure generated by the model. Then for better illustration, we switch the order in the tuple as below, by applying graph concepts, we have the first two as the 'node' positions and the last as the 'edge' position:

$$(a^{p_1}, k_1, a_1) \Rightarrow (\underbrace{a^{p_1}, a_1}_{nodes}, \underbrace{k_1}_{edge}) \tag{13}$$

where the order of two nodes is defined by the reasoning LLM.

Now, we can write a basic reasoning step as:

$$\text{Step}_{ij} = [\text{node}_i, \text{edge}_{ij}, \text{node}_j] \tag{14}$$

where $\text{node}_i$ is the starting node representing either a question, a sub-question, or an intermediate response, $\text{node}_j$ is the resulting node from $\text{node}_i$, and connected by $\text{edge}_{ij}$, which serves as the reasoning operation or sub-question.

Specifically, for the initial input query $x_q$, we denote the node as nodeRaw; for the final answer $a$, we denote as nodeResult. All other steps in the middle are the reasoning process, with a clearly defined structure. The final graph structure includes all reasoning steps from query

---

[3]Please find details of few-shot learning in Appendix.

$x^q$ to the final answer $a$ as nodes ($v_i$) and their dependencies as edges ($e_{ij}$): The reasoning process from query $q$ to the final answer $a$ can be finalized as a directed graph structure

$$\mathcal{G}^q = (\mathcal{V}, \mathcal{E}), \tag{15}$$

where

$$\mathcal{V} = \{\text{nodeRaw}, \text{node}_1, \dots, \text{nodeResult}\} = \{v_0, v_1, \dots\} \tag{16}$$

and the edges are expressed as:

$$\mathcal{E} = \{e_{ij(1)}, e_{ij(2)} \dots\} \tag{17}$$

where $e$ stands for edge and $\{e_{ij} \mid \text{edge}_{ij} : \text{node}_i \to \text{node}_j\}$, $e_{ij}$ represents reasoning operations or dependencies between nodes. (start with index '1' since we assume '0' is the nodeRaw). The graph-based structure captures the full reasoning topology, including branching, dependencies, and multi-step interactions, which allows for a better reflection of the relationships between intermediate steps.

Now, from a graph concept, the reasoning steps combined with Eq. 14 are formalized as below:

$$S = \{\text{Step}_{ij} \mid \text{Step}_{ij} = [v_i, e_{ij}, v_j] v_i, v_j \in \mathcal{V}, e_j \in \mathcal{E}\} \tag{18}$$

where each triplet represents a logical transition between reasoning steps. Note that for complex reasoning, the final answer does not necessarily rely on all of the reasoning steps. For example, when being asked about "If it is currently summer in Australia, what season is it in Canada?" in the reasoning chain, some of the LLM might delve into 'what causes the season differences, ' which is a redundant step in concise reasoning.

## C More Studies for Reasoning Graph Edit Distance

### C.1 The case behind high $c_{\text{sub}}$.

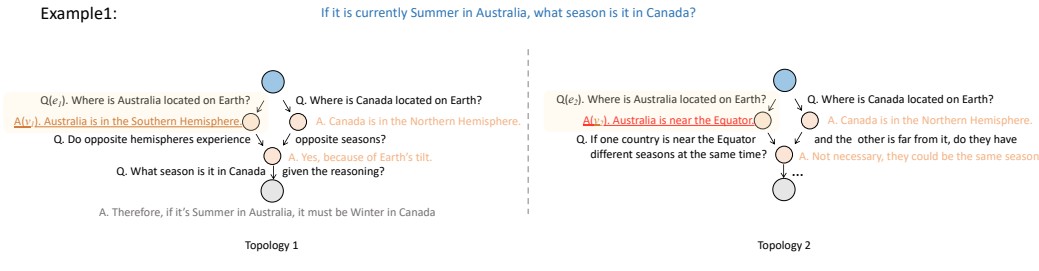

Figure 5: The example of the cause of substitution cost: same sub-question, divergence answers.

### C.2 The complete equation of deletion/insertion cost

Similarly to Eq. equation 5 for nodes, we define the deletion cost for an edge $e_i \in \mathcal{E}_1$ in graph $\mathcal{G}_1 = (\mathcal{V}_1, \mathcal{E}_1)$ relative to the target graph $\mathcal{G}_2 = (\mathcal{V}_2, \mathcal{E}_2)$ as:

$$c_{\text{del}}(e_i) = \frac{1}{2} \{ \underbrace{\max_{e_j \in \mathcal{E}_2} \cos\left(\mathcal{L}(e_i), \mathcal{L}(e_j)\right)}_{\text{Cross-graph Matching Term}} + 1 - \underbrace{\frac{1}{|\mathcal{E}_1| - 1} \sum_{\substack{e_k \in \mathcal{E}_1 \\ e_k \neq e_i}} \cos\left(\mathcal{L}(e_i), \mathcal{L}(e_k)\right)}_{\text{Internal Uniqueness Term}} \}. \tag{19}$$

The Eq. equation 19 jointly considers the cross-graph matching cost and the internal uniqueness cost when deleting edge $e_i$.

- **Cross-graph Matching Term:**

$$\max_{e_j \in \mathcal{E}_2} \cos \left( \mathcal{L}(e_i), \mathcal{L}(e_j) \right)$$

measures the highest semantic similarity between the edge $e_i$ in $\mathcal{G}_1$ and any edge $e_j$ in the target graph $\mathcal{G}_2$. If $e_i$ has a very similar counterpart in $\mathcal{E}_2$, deleting it would remove important alignment information, leading to a higher deletion cost.

- **Internal Uniqueness Term:**

$$1 - \frac{1}{|\mathcal{E}_1| - 1} \sum_{\substack{e_k \in \mathcal{E}_1 \\ e_k \neq e_i}} \cos \left( \mathcal{L}(e_i), \mathcal{L}(e_k) \right)$$

calculates how unique edge $e_i$ is within its own graph $\mathcal{G}_1$. If $e_i$ is very similar (i.e., redundant) to many other edges, the average similarity will be high, making this term low; in other words, deleting a redundant edge is less costly.

Thus, the overall deletion cost for edge $e_i$ is the average of these two terms.

## D  Related Work

In this section, we review the related work in the research domains of uncertainty quantification (UQ) for large language models (LLMs) and methods for explanation-based UQ, with a focus on reasoning processes.

### D.1  UQ for LLM

**White-box Approaches**  A significant body of research has focused on performing UQ for LLMs by inducing the models to output their uncertainty along with their responses Kadavath et al. (2022); Lin et al. (2022); Mielke et al. (2020); Tian et al. (2023). These methods often rely on token-level probabilities to train or fine-tune models for predicting uncertainty. While effective, these approaches require full access to the model's structure and weights, which is impractical for black-box or commercial LLMs. For example, supervised methods such as those in Kadavath et al. (2022) estimate uncertainty using logits and ground truth labels but are computationally expensive and resource-intensive.

**Black-box Approaches**  Another line of work estimates uncertainty directly at the response level using semantic entropy Kuhn et al. (2023). While this method avoids token-level dependencies, it still relies on access to token probabilities, limiting its applicability in black-box settings. To address these limitations, researchers have proposed lightweight black-box methods that analyze response inconsistencies. For instance, Lin et al. (2023) uses graph Laplacian eigenvalues as an uncertainty indicator, while Chen & Mueller (2023) computes confidence scores from generated outputs to identify speculative or unreliable answers. However, these approaches primarily focus on semantic-level analysis and neglect the logical structure underlying reasoning processes. Moreover, methods like Lin et al. (2023) average entailment probabilities without considering directional information in reasoning paths.

Our work is agnostic to the white box or black box since it leverages the generated explanations as a proxy to measure the reasoning uncertainty as in Figure. 1. This enables a more nuanced and interpretable assessment of uncertainty in reasoning processes.

### D.2  UQ for LLM Explanation

Explanation-based UQ focuses on assessing the reliability of natural language explanations (NLEs) generated by LLMs by either prompting models to express confidence in their explanations or analyzing consistency across multiple outputs under varying conditions Tanneru et al. (2024); Yadkori et al. (2024). While these methods provide insights into explanation

robustness, they treat explanations to a question as unstructured text representation, which lacks structural information and fails to capture inconsistencies or leaps in logic. In contrast, our work explicitly leverages well-structured reasoning topologies to enhance the UQ process for explanations. This structured representation enables us to assess explanation uncertainties at a finer granularity within complex reasoning paths.

# E   Details of the GeoQA Dataset

The GeoQA dataset is designed to evaluate the reasoning capabilities of large language models (LLMs) on conditional geographical questions, emphasizing the comparative reasoning topology of their responses. By anchoring specific knowledge within conditional constraints and requiring models to infer results or solutions, GeoQA enables an in-depth analysis of the reasoning paths taken by LLMs. The dataset spans 20 categories, covering diverse geographical topics such as climate, biome, tectonic plates, continental drift, altitude, sea level, desertification, urbanization, demography, population density, ocean currents, river basin, watershed, mountain range, volcano, earthquake, glacier, permafrost, and monsoon. Each question is crafted to test multi-step reasoning, integration of domain-specific knowledge, and the ability to navigate complex cause-effect relationships, making GeoQA a unique and challenging benchmark for geographical reasoning.

# F   Empirical Study on the Choice of Number for Generations

It is a basis setup that we need to query LLM $\mathcal{M}$ with a query $x^q$ for $k$ times and collect a set of explanations to perform the NLE uncertainty measure. We have conducted a survey on related literature and found there is no standard definition or setting, so we conducted a preliminary study on the number of responses and tried to find the most suitable one (since the larger the response is, the more computationally expensive it will be for later evaluation).

# G   Details of Baseline methods

## G.1   CoTA.

Chain-of-Thought Agreement (CoTA) evaluates the agreement between two Chain-of-Thought (CoT) explanations generated for the same query. Each CoT explanation consists of a sequence of reasoning steps, denoted as:

$$CoT_a = \{s_{a1}, s_{a2}, \ldots, s_{aN_a}\}, \quad CoT_b = \{s_{b1}, s_{b2}, \ldots, s_{bN_b}\}.$$

The CoTA metric quantifies agreement between the two CoT explanations by calculating the maximum semantic alignment for each step in $CoT_a$ with steps in $CoT_b$, and vice versa. Formally, CoTA is defined as:

$$\text{CoTA}(CoT_a, CoT_b) = \frac{1}{N_a + N_b} \left( \sum_{i=1}^{N_a} \max_{j=1,\ldots,N_b} E(s_{ai}, s_{bj}) \right. \tag{20}$$
$$\left. + \sum_{j=1}^{N_b} \max_{i=1,\ldots,N_a} E(s_{bj}, s_{ai}) \right)$$

where $N_a$ and $N_b$ are the number of steps in $CoT_a$ and $CoT_b$, respectively.

The entailment function $E(s_i, s_j)$ measures the semantic agreement between two reasoning steps using a Natural Language Inference (NLI) model. It is defined as:

$$E(s_i, s_j) = \begin{cases} 1, & \text{if } s_i \text{ entails } s_j, \\ 0, & \text{otherwise.} \end{cases}$$

Table 3: Examples of GeoQA dataset.

| Question Type | Type Explain | Example Questions | Analysis |
|---|---|---|---|
| Glacier | These questions explore glacial movement, erosion, and the impact of climate change on ice dynamics. | ① : If glaciers carve striations into bedrock, what do these scratches indicate about past movements? | Striations indicate past movement direction, requiring cause-and-effect analysis, while glacier mass loss demands understanding the imbalance between accumulation and ablation. |
| | | ② : If a glacier loses mass due to melting and sublimation exceeding accumulation, what process is occurring? | |
| Earthquake | These questions focus on seismic wave behavior, fault activity, and earthquake detection. | ① : If seismic waves are recorded by a network of seismographs, which method is used to pinpoint the origin of the disturbance? | Triangulation requires reasoning through wave arrival times, while P-wave detection relies on comparing wave speeds and impact to explain early warning systems. |
| | | ② : If earthquake early warning systems rely on detecting the initial P-waves, which characteristic of these waves makes this feasible? | |
| permafrost | These questions examine permafrost thawing, climate feedback loops, and seasonal variations. | ① : If permafrost thaws due to rising temperatures, releasing trapped methane, what global issue does this exacerbate? | Thawing permafrost and methane release involve feedback loops, while active layer variations require analyzing environmental factors like temperature and insulation. |
| | | ② : If the active layer above permafrost varies in thickness seasonally, what factors influence its depth? | |
| Monsoon | These questions cover seasonal wind shifts, monsoon patterns, and storm formation. | ① : If the East Asian monsoon affects countries like China and Japan, what two seasons does it primarily influence? | The East Asian monsoon's impact on seasons involves reasoning about wind shifts, while monsoon depressions require linking low-pressure systems to storm formation. |
| | | ② : If monsoon depressions form in the Bay of Bengal, what weather events might they trigger upon landfall? | |

The entailment model employs pre-trained NLI models, such as DeBERTa (He et al., 2020), fine-tuned for evaluating entailment relationships between statements. This binary scoring avoids dependency on confidence calibration, and we take the threshold as 0.7 to provide the binary cut.

## G.2 Embed-UQ.

Embed-UQ measures uncertainty by embedding natural language explanations into a semantic space and computing the variance of pairwise distances. Given a query $x^q$, let $\{a_1^e, a_2^e, \ldots, a_k^e\}$ represent $k$ explanations generated by the model. Using an embedding function $\mathcal{L}(\cdot)$, each explanation $a_i^e$ is mapped to a high-dimensional embedding:

$$\mathbf{h}_i = \mathcal{L}(a_i^e).$$

The pairwise distances between embeddings are computed as:

$$d_{ij} = \|\mathbf{h}_i - \mathbf{h}_j\|,$$

where $d_{ij}$ represents the distance between explanations $a_i^e$ and $a_j^e$. The uncertainty is then quantified as the variance of the distance matrix $D$:

$$\mathcal{U}(x^q) = \text{Var}(D),$$

where $D = \{d_{ij} \mid 1 \leq i, j \leq k\}$.

## G.3 Entail-UQ.

Entail-UQ modifies the distance computation in Embed-UQ by using an entailment-based similarity measure instead of embedding distances. Given the same set of explanations $\{a_1^e, a_2^e, \ldots, a_k^e\}$, an entailment model computes the similarity between two explanations $a_i^e$ and $a_j^e$ as:

$$s_{ij} = E(a_i^e, a_j^e),$$

where $E(\cdot, \cdot)$ is the entailment function that outputs a similarity score between 0 and 1. The dissimilarity is then defined as $1 - s_{ij}$, and the uncertainty is computed as the variance of the dissimilarity matrix $S$:

$$\mathcal{U}(x^q) = \text{Var}(1 - S),$$

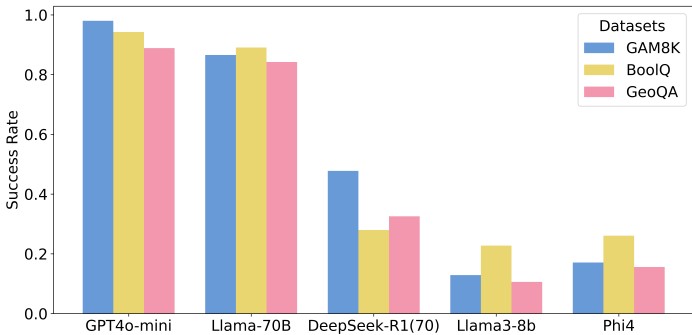

Figure 6: The model's success rate to generate legitimate reasoning topology. It is calculated by the percentage of LLMs successfully generating the reasoning path from the nodeRaw to nodeResult following the few-shot prompt. It can be witnessed that, generally the GeoQA is a harder one to generate due to the conditional question types, which are rarely seen in the LLM training tasks.

where $S = \{s_{ij} \mid 1 \leq i, j \leq k\}$. Similarly to CoTA, in our implementation, we adopt DeBERTa (He et al., 2020) as the model to generate entailment logit and pass through a layer of softmax to transform it into probabilities.

### G.4 NLI-logit-UQ.

NLI-logit-UQ is a variant of Entail-UQ that directly utilizes the raw logits from a Natural Language Inference (NLI) model to measure uncertainty, without applying the softmax operation. Given the set of explanations $\{a_1^e, a_2^e, \ldots, a_k^e\}$, the NLI model computes a raw logit score for each pair of explanations:

$$l_{ij} = \text{NLI}(a_i^e, a_j^e),$$

where $\text{NLI}(\cdot, \cdot)$ outputs a logit value representing the degree of entailment between $a_i^e$ and $a_j^e$.

In contrast to Entail-UQ, where these logits are passed through a softmax layer to obtain probabilities, the NLI-logit-UQ method uses the raw logits directly. The dissimilarity between explanations can be defined as:

$$d_{ij} = 1 - l_{ij},$$

or more generally, the uncertainty is computed as the variance of the raw logit values:

$$\mathcal{U}(x^q) = \text{Var}(\{l_{ij} \mid 1 \leq i, j \leq k\}).$$

## H   Research Methods

### H.1   Part One

The final full version of the graph edit distance is as below:

$$\text{GED}(\mathcal{G}_1, \mathcal{G}_2) = C_{\text{sub.}}(\mathcal{P}) + C_{\text{del.}}(\mathcal{V}_1, \mathcal{E}_1, \mathcal{P}) \tag{21}$$

where $\mathcal{P}$ represents the optimal matchings for nodes ($\mathcal{P}_v$) and edges ($\mathcal{P}_e$), computed using an algorithm such as the Hungarian algorithm (Hamuda et al., 2018). The term $C_{\text{sub.}}(\mathcal{P})$ accounts for substitution costs, defined as:

$$C_{\text{sub.}}(\mathcal{P}) = \sum_{(v_i, v_j) \in \mathcal{P}_v} c(v_i, v_j) + \sum_{(e_i, e_j) \in \mathcal{P}_e} c(e_i, e_j), \tag{22}$$

where $c(v_i, v_j)$ and $c(e_i, e_j)$ represent the node and edge substitution costs, respectively. The term $C_{\text{del.}}(\mathcal{V}_1, \mathcal{E}_1, \mathcal{P})$ captures the deletion costs for nodes and edges in $\mathcal{G}_1$ that are not matched, given by:

$$C_{\text{del.}}(\mathcal{P}) = C_{\text{del.}}(\mathcal{V}_1, \mathcal{E}_1, \mathcal{P}) = \sum_{v_i \in \mathcal{V}_1 \setminus \mathcal{P}_v} c_{\text{del.}}(v_i) + \sum_{e_i \in \mathcal{E}_1 \setminus \mathcal{P}_e} c_{\text{del.}}(e_i). \tag{23}$$

Here, $c_{\text{del.}}(v_i)$ and $c_{\text{del.}}(e_i)$ denote the deletion costs of unmatched nodes and edges, respectively. This formulation quantifies the total cost required to align the two reasoning structures by summing the substitution costs of matched components and the deletion costs of unmatched ones.

### H.2 Algorithm for Detection of Redundant Nodes and Dead Branches

To identify redundant nodes and branches of dead nodes in the reasoning topology $S = \{[v_i, v_j, e_{ij}] \mid v_i, v_j \in \mathcal{V}, e_{ij} \in \mathcal{E}\}$, we first define the outgoing edges of a node $v_k$ as: $\text{Out}(v_k) = \{v_j \mid [v_k, v_j, e_{kj}] \in S\}$ Then A node $v_k$ is considered redundant if it has no outgoing edges and is not the final node:

$$\text{Out}(v_k) = \varnothing \quad \text{and} \quad v_k \neq \text{NodeResult}. \tag{24}$$

Then, we compute the set of valid paths, $\mathcal{P}_{\text{valid}}$, connecting NodeRaw to NodeResult using DFS. A node $v_k \in \mathcal{V}$ is redundant if it does not appear in any valid path:

$$v_k \notin \bigcup_{[v_i, v_j, e_{ij}] \in \mathcal{P}_{\text{valid}}} \{v_i, v_j\}. \tag{25}$$

In order to more efficiently detect branches of dead nodes (not contributing to the whole reasoning path), let $v_k$ and its parent $v_p$ satisfy:

$$\text{Out}(v_p) = \{v_k\}, \quad \text{and} \quad v_p, v_k \notin \bigcup_{[v_i, v_j, e_{ij}] \in \mathcal{P}_{\text{valid}}} \{v_i, v_j\}.$$

In this case, $v_p$ and $v_k$ form a dead branch, as neither contributes to any reasoning path leading to NodeResult. And finally, the redundancy rate is computed as:

$$redun.(a_i^e) = \frac{|\mathcal{V}_{\text{redundant}}|}{|\mathcal{V}|},$$

. It allows to systematically identify nodes and branches that do not contribute to the reasoning process, providing insights into the inefficiencies in the model's reasoning topology and we can analyze to understand where we can improve in the model training or fine-tuning process.

## I Experimental Details

### I.1 Details of the dataset

- **GSM8K (Cobbe et al., 2021)**: This dataset contains 8,000 high-quality math word problems, designed to evaluate LLMs' ability to perform arithmetic reasoning. It is a standard benchmark for testing the numerical and reasoning capabilities of LLMs.
- **BoolQ (Clark et al., 2019)**: BoolQ is a yes/no question-answering dataset derived from naturalistic information-seeking questions. The dataset is used to evaluate the ability of LLMs to reason logically over textual evidence and produce accurate binary answers.
- **GeoQA**: GeoQA is a self-constructed dataset designed to evaluate the reasoning capabilities of LLMs in conditional questions. With 20 categories, including climate and tectonic processes, its tasks require inference of specific results from given conditions. GeoQA emphasizes multi-step reasoning and domain-specific integration, making it a challenging benchmark (details in the Appendix E).

### I.2 The calculation of the faithfulness core

In our experiments, we utilize a strategy called **Early Answering** to measure the faithfulness of the reasoning paths $a^e = \{s_1, s_2, \ldots, s_n\}$, which are generated by the LLM for a given query $x^q$. This strategy involves truncating the reasoning steps $a^e$ progressively and prompting the model to answer the query $x^q$ combined with the partial reasoning path $\{s_1, s_2, \ldots, s_k\}$, where $k \in \{1, 2, \ldots, n\}$. For example, instead of providing the entire reasoning $x^q + s_1 + s_2 + \cdots + s_n$, the model is prompted to answer using only $x^q + s_1$, $x^q + s_1 + s_2$, and so on, until the full reasoning path is reached.

The Early Answering process evaluates how often the LLM's responses, derived from the partial reasoning path $\{s_1, s_2, \ldots, s_k\}$, match the final answer $a$ generated using the complete reasoning $x^q + s_1 + s_2 + \cdots + s_n$. This evaluation reflects the faithfulness of the reasoning path: if the model consistently reaches the correct answer $a$ with partial reasoning, it may indicate that the reasoning steps are unnecessary (post-hoc). Conversely, a lower match rate suggests that the intermediate reasoning steps are essential for arriving at the correct final answer, thereby indicating greater faithfulness.

We quantify the faithfulness $V_{faith}$ using the following equation:

$$V_{\text{faith}} = 1 - \underbrace{\frac{1}{n} \sum_{k=1}^{n} \mathbb{I}\left(f(x^q + \{s_1, s_2, \ldots, s_k\}) = a\right)}_{\text{un-faithfulness}} \tag{26}$$

where $n$ represents the total number of reasoning steps in $a^e$, $f(x^q + \{s_1, s_2, \ldots, s_k\})$ is the LLM's output when prompted with the query $x^q$ and the partial reasoning path $\{s_1, s_2, \ldots, s_k\}$, $a$ denotes the final answer generated using the complete reasoning path, and $\mathbb{I}(\cdot)$ is an indicator function that equals 1 if the condition is true (i.e., the partial reasoning output matches the final answer), and 0 otherwise. The $V_{\text{faith}}$ is calculated by $1 - \text{un-faithfulness}$, and the unfaithfulness means: how much the model's final answer $a$ (Not) depends on the intermediate reasoning steps, since by removing sub-steps, it still reaches same answer.

A high faithfulness score indicates that the model's final answer is more dependent on intermediate reasoning steps, suggesting that the reasoning is not post-hoc, and this faithfully reflects that the logical steps are required to derive the answer, by this high faithfulness, the UQ measure should align with it, in other words, a UQ method is good if it derives a lower uncertainty when the faithfulness is high, and vise versa.

## J  Evaluation Metrics Calculation

Let $\{(x_i, y_i)\}_{i=1}^{n}$ be a set of paired samples, where $x_i$ refers to the *uncertainty value* and $y_i$ refers to the *ground-truth faithfulness*. All three correlation metrics produce values in the range $[-1, 1]$:

- $\rho = 1$ means a perfect **positive** correlation,

- $\rho = -1$ means a perfect **negative** correlation,

- $\rho = 0$ means **no** correlation at all.

**Note:** In our setting, a **more negative** correlation is preferred because we expect *higher* uncertainty to correspond to *lower* faithfulness (and vice versa).

**1. Pearson Correlation Coefficient (PCC)**

$$\rho_{X,Y} = \frac{\sum_{i=1}^{n}(x_i - \bar{x})(y_i - \bar{y})}{\sqrt{\sum_{i=1}^{n}(x_i - \bar{x})^2}\sqrt{\sum_{i=1}^{n}(y_i - \bar{y})^2}},$$

where $\bar{x} = \dfrac{1}{n}\sum_{i=1}^{n} x_i$ and $\bar{y} = \dfrac{1}{n}\sum_{i=1}^{n} y_i$ are the sample means of $X$ and $Y$, respectively. This measures the *linear* correlation between $x$ and $y$.

## 2. Spearman Rank Correlation (SRC)

First, convert each $x_i$ and $y_i$ into their respective ranks, denoted $\mathrm{rank}(x_i)$ and $\mathrm{rank}(y_i)$. Then define

$$d_i = \mathrm{rank}(x_i) - \mathrm{rank}(y_i).$$

The Spearman rank correlation coefficient $r_s$ is given by

$$r_s = 1 - \frac{6\sum_{i=1}^{n} d_i^2}{n(n^2 - 1)}.$$

Spearman's method captures the *monotonic* relationship between the two variables, based on their ranks rather than their absolute values.

## 3. Kendall Rank Correlation (KRC)

For each pair $(i, j)$ with $i < j$, define

$$\mathrm{sgn}(x_j - x_i) = \begin{cases} 1 & \text{if } x_j > x_i, \\ -1 & \text{if } x_j < x_i, \\ 0 & \text{otherwise,} \end{cases} \qquad \mathrm{sgn}(y_j - y_i) = \begin{cases} 1 & \text{if } y_j > y_i, \\ -1 & \text{if } y_j < y_i, \\ 0 & \text{otherwise.} \end{cases}$$

Then the Kendall rank correlation $\tau$ can be computed as

$$\tau = \frac{2}{n(n-1)} \sum_{i<j} \mathrm{sgn}(x_j - x_i)\,\mathrm{sgn}(y_j - y_i).$$

Equivalently, one may count the number of *concordant* pairs $n_c$ and *discordant* pairs $n_d$, yielding

$$\tau = \frac{n_c - n_d}{\frac{1}{2}n(n-1)}.$$

Kendall's $\tau$ also measures the degree of *ordinal association* between two variables, similar to Spearman's rank correlation but with a different counting approach.

| Methods | GPT4o-mini | | | DeepSeek-R1 | | | Llama-3.3-70B | | | Llama3-8b | | | Phi4 | | |
|---|---|---|---|---|---|---|---|---|---|---|---|---|---|---|---|
| | PCC($\downarrow$) | SRC($\downarrow$) | KR($\downarrow$) | PCC($\downarrow$) | SRC($\downarrow$) | KR($\downarrow$) | PCC($\downarrow$) | SRC($\downarrow$) | KR($\downarrow$) | PCC($\downarrow$) | SRC($\downarrow$) | KR($\downarrow$) | PCC($\downarrow$) | SRC($\downarrow$) | KR($\downarrow$) |
| **Dataset: GeoQA** | | | | | | | | | | | | | | | |
| CoTA | -0.18 | -0.09 | -0.12 | -0.14 | -0.15 | 0.10 | -0.03 | -0.04 | -0.03 | 0.01 | 0.01 | 0.01 | -0.02 | -0.04 | -0.02 |
| Embed-UQ | -0.11 | -0.10 | -0.10 | 0.71 | 0.69 | 0.50 | -0.01 | 0.01 | 0.01 | 0.08 | 0.09 | 0.06 | -0.01 | -0.02 | -0.01 |
| Entail-UQ | -0.12 | -0.05 | -0.05 | 0.65 | 0.64 | 0.46 | **-0.25** | **-0.23** | **-0.15** | -0.05 | -0.06 | -0.04 | -0.05 | -0.04 | -0.03 |
| NLI-logit-UQ | 0.08 | 0.12 | 0.10 | 0.72 | 0.70 | 0.51 | -0.05 | -0.05 | -0.03 | 0.09 | 0.10 | 0.07 | -0.01 | -0.02 | -0.01 |
| Ours | **-0.22** | **-0.34** | **-0.31** | **-0.29** | **-0.31** | **-0.19** | -0.11 | -0.14 | -0.08 | **-0.19** | **-0.11** | **-0.09** | -0.03 | -0.04 | -0.03 |

Table 4: Comparison of our methods with different baselines on various datasets and large language models. Cont. for GeoQA dataset.

# K   Time Complexity Analysis

Below we provide an explanation of the time complexity analysis using our **Reason-GED**:

**Let:**   $\mathcal{G}_1, \mathcal{G}_2, \ldots, \mathcal{G}_n$ be the $n$ reasoning graphs generated for a query $x_q$. For each pair of graphs $\mathcal{G}_i$ and $\mathcal{G}_j$, the graph edit distance is computed as: $d_{ij} = \mathrm{GED}(\mathcal{G}_i, \mathcal{G}_j)$.

Assume the cost to compute the graph edit distance between two reasoning graphs is $T_{\mathrm{GED}}$. In our approach, we compute this distance for every pair, the amount of such pairs are:

$$\binom{n}{2} = \frac{n(n-1)}{2}. \tag{27}$$

Thus, the overall time complexity for computing all pairwise distances is:

$$O\left(\frac{n(n-1)}{2} \cdot T_{\text{GED}}\right) = O\left(n^2 \cdot T_{\text{GED}}\right). \tag{28}$$

If each reasoning graph has $m$ nodes and the matching step (e.g., using the Hungarian algorithm) requires $O(m^3)$ time, then: $T_{\text{GED}} = O(m^3)$, so, the total time complexity for estimating the uncertainty for one query is: $O\left(n^2 \cdot m^3\right)$. If $m$ is bounded or considered a constant, the complexity simplifies to: $O\left(n^2\right)$

This concludes the mathematical analysis of the time complexity for uncertainty estimation per question when we have $n$ responses.

## L   Prompt Template & Few Shot Examples

Here we introduce details of the prompt template used in this paper as well as some few-shot examples to guide the LLMs to follow the elicitation process.

---

**The Prompt Template to Elicit Knowledge Points.**

- - - - - - - - - - - - - - - - - - - - - - - - - - - - - - - - - - - - - - - - - - - - - - - - - - - - - - - -

**System Description:** You are a helpful assistant to do the following: Given a question, you should reflect and come up with the sufficient knowledge that you need to solve this question. Two standards: sufficient and concise. And you should respond with numbered points.

**Task Description:** Given a question: { question_i }. Please provide a response following system requirements and learning the format from the example: { few_shot_example }.

**Example1:**

Question: If it is currently summer in Australia, what season is it in Canada?

Expected Response (For required knowledge):

1. Where is Australia located on Earth?

2. Where is Canada located on Earth?

3. What is the geographical relationship between Australia and Canada?

4. How does the tilt of the Earth affect seasons?

**Examples ...**

**Output:** {Placeholder}

---

**The Prompt Template to Express Reasoning Path.**

- - - - - - - - - - - - - - - - - - - - - - - - - - - - - - - - - - - - - - - - - - - - - - - - - - - - - - - -

**System Description:** You are a reasoning assistant, you will see some Edge-Node pairs, which stands for the Q-A pairs, try to find a reasoning path based on these Q-A pairs that solves the question.

**Task Description:** Given a { question }. Please learn how it is reasoned from the example: Reason_Path_Example. Now give the reasoning path for {q_a}.

**Constraints:**

1. NodeRaw and NodeResult are nominal term, NodeRaw stands for Question itself and NodeResult stands for the End of reasoning. ;

2. When reason to the conclusion, there should be an added: ResultNode and ResultEdge as: [Nodex, NodeResult, ResultEdge];

3. [NodeRaw, Node0, Edge0]: indicates NodeRaw is connected with Node0 by Edge0. [Nodex, NodeResult, ResultEdge]: indicates Nodex is connected with NodeResult by ResultEdge.;

**Example1:**

Question: If it is currently summer in Australia, what season is it in Canada?

Edge0: Where is Australia located on Earth?, Node0: Australia in the Southern Hemisphere.;

Edge1: Where is Canada located on Earth?, Node1: Canada is located in the Northern Hemisphere.;

Edge2: What is the geographical relationship between Australia and Canada?, Node2: Australia and Canada are in the opposite hemisphere.;

Edge3: How does the tilt of the Earth affect seasons?, Node3: Opposite hemispheres experience opposite seasons because of the Earth's tilt.;

**A Possible Output:**

Structure: [NodeRaw, Node0, Edge0], [NodeRaw, Node1, Edge1], [Node0, Node2, Edge2], [Node1, Node2, Edge2], [Node2, Node3, Edge3], [Node3, NodeResult, ResultEdge]; ResultEdge: It is summer in Canada.;

**Output:** {Placeholder}

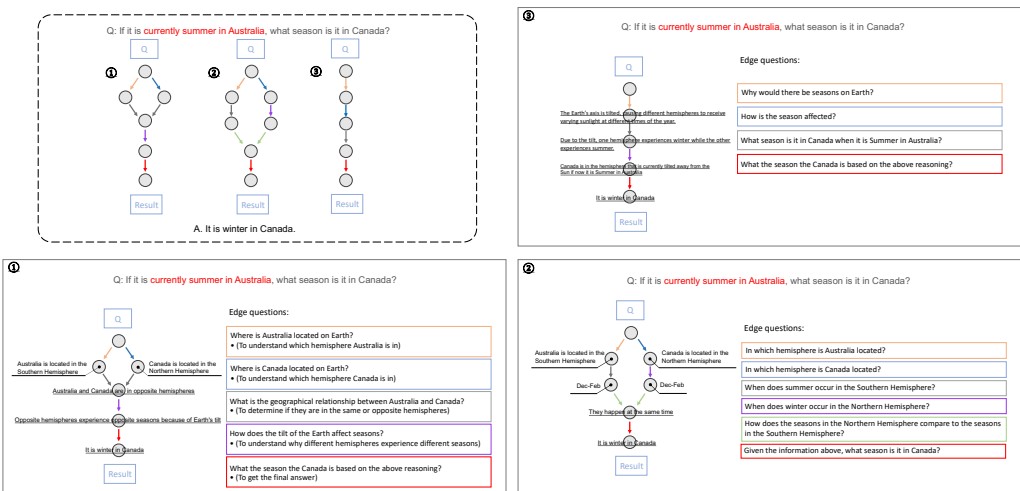

Figure 7: The Example of of same question but different reasoning path and leading to the same answer: 'If it is currently summer in Australia, what season is it in Canada?' .

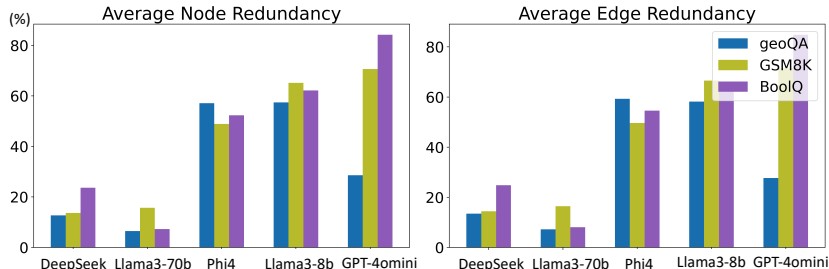

Figure 8: The complete version of LLM's reasoning redundancy on three datasets: node and edge.

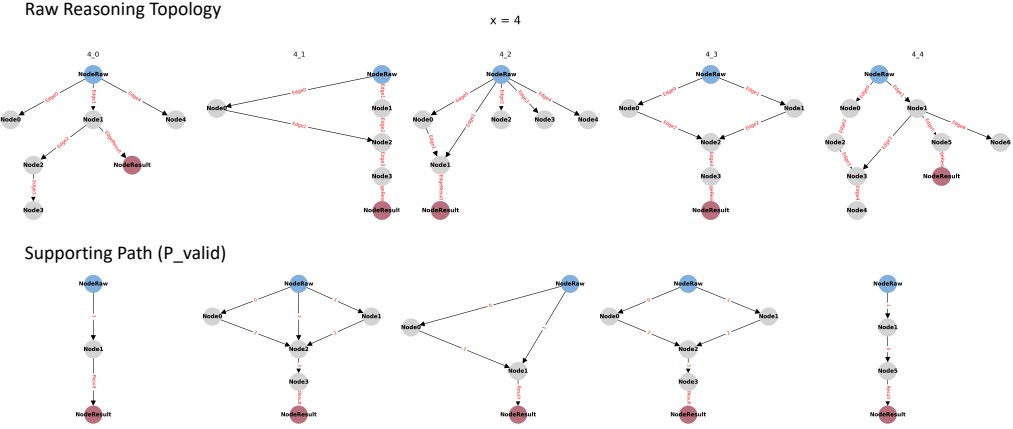

Figure 9: The Example of the redundancy for LLMs (GPT4o-mini)

