# OpenReview forum: "Understanding the Uncertainty of LLM Explanations: A Perspective Based on Reasoning Topology"
_colmweb.org/COLM/2025/Conference — COLM 2025_

### Official Review · Reviewer_GC6U · 2025-05-06

**Rating:** 7
**Confidence:** 2
**Ethics Flag:** 1

**Summary:**

The authors introduce Topo‑UQ, a model‑agnostic framework for estimating uncertainty in large‑language‑model (LLM) explanations. Instead of treating each explanation as a flat text, Topo‑UQ first decomposes the model’s chain‑of‑thought into a reasoning graph whose nodes are the sub‑questions and intermediate answers produced by the LLM. Uncertainty is then quantified with Reason‑GED, a graph‑edit‑distance that blends structural edits (insertions, deletions, substitutions of nodes / edges) with semantic similarity between the texts attached to those elements. Larger variance across several independently generated graphs indicates higher epistemic uncertainty. A complementary metric, reasoning redundancy, flags superfluous steps that do not contribute to the final answer.

Experiments on GSM8K (math), BoolQ (fact verification) and GeoQA (geospatial queries) with five open‑source and proprietary LLMs show that Topo‑UQ’s scores correlate more strongly with answer reliability than four existing text‑only baselines.
Because Topo‑UQ needs only the model’s textual outputs, it works with black‑box APIs, yet its effectiveness depends on the LLM’s ability to follow the multi‑step prompting protocol.

**Reasons To Accept:**

Strength:

- The paper is clear and easy to follow (even for a non-specialist in Uncertainty Quantification methods like me).
- Turning each explanation into a reasoning graph is a fresh perspective that surfaces structural sources of uncertainty ignored by flat‑text methods.
- Blending graph‑edit operations with sentence‑level similarity lets the framework detect subtle divergence that pure text metrics miss.
- The additional “reasoning‑redundancy” indicator pinpoints superfluous steps, offering a concrete lever for trimming inference cost.
- Tests on three benchmark datasets and five diverse LLMs show consistently stronger correlations with answer reliability than four competitive baselines (CoTA, Embed‑UQ, Entail‑UQ, NLI‑logit).

**Reasons To Reject:**

Weakness:

- Generating multiple chains, building reasoning graphs, and computing pairwise Reason‑GED scores scales quadratically with the number of samples, making the approach expensive for real‑time or large‑batch settings. The authors don't give the computational cost of their method, which they should note. It would also be interesting to have figures on the computational cost of each baseline method, in order to better assess the Topo‑UQ method in relation to the others.
- The study reports automatic correlations but omits user studies that would show whether the uncertainty scores actually improve human confidence.
- The authors state in the limitations that less instruction‑tuned LLMs could undermine the protocol, yet they provide no empirical evidence, but we need to see that. An appendix evaluating a lightweight model such as Llama3‑1B would clarify how sharply Topo‑UQ’s performance diverges from baselines (CoTA, Embed‑UQ, Entail‑UQ, NLI‑logit) under low‑capacity settings.

---

> ### Author Response · Authors · 2025-06-03
> **Reply to Reviewer GC6U**
>
> **We are thankful for the reviewer's suggestions. The responses to the weakness are in the following:**
>
> > Generating multiple chains, building reasoning graphs, and computing pairwise Reason‑GED scores, etc, making the approach expensive for real‑time or large‑batch settings.
>
> The uncertainty quantification of LLMs can be divided into two directions (black box, white box) based on the accessibility to the model. For black box models, without access to the model's log probability and generated logits, it is difficult to provide efficient quantification. Thus, it has been a standard to prompt multiple times for UQ analysis [1][2][3]. Based on this, since we delve deep into the reasoning process, it is required to extract reasoning topologies for further analysis.
>
> [1] Lin Z,  et al. Generating with Confidence: Uncertainty Quantification for Black-box Large Language Models[J]. Transactions on Machine Learning Research.
>
> [2] Liu X, et al. Uncertainty quantification and confidence calibration in large language models: A survey[J]. arXiv preprint arXiv:2503.15850, 2025.
>
> [3] Ling C, et al. Uncertainty Quantification for In-Context Learning of Large Language Models, Proceedings of the 2024 Conference of the North American Chapter of the Association for Computational Linguistics.
>
>
> > It would also be interesting to see the computational cost of each baseline method, in order to better assess the Topo‑UQ method in relation to the others.
>
> Thanks for the advice, we have designed an extra experiment focusing on assessing the computational cost for each of the baseline methods. The experiment was conducted on the same response set collected from the BoolQ question set, and covers five methods: CoTA, Embed-UQ, Entail-UQ, NLI-logit, and our method (Topo-UQ). Total 1280 responses, 10 times for each question.
>
> |   Run    | Topo-UQ (Ours) |  CoTA  | NLI-logit | Embed-UQ | Entail-UQ |
> |:--------:|:------------:|:------:|:---------:|:--------:|:---------:|
> |    1     |    0.1039    | 0.2249 |   0.0338  |  0.0165  |   0.1478  |
> |    2     |    0.1034    | 0.2246 |   0.0334  |  0.0165  |   0.1459  |
> |    3     |    0.1046    | 0.2204 |   0.0329  |  0.0163  |   0.1449  |
> |    4     |    0.1059    | 0.2210 |   0.0345  |  0.0162  |   0.1460  |
> |    5     |    0.1044    | 0.2271 |   0.0339  |  0.0167  |   0.1483  |
> | **Average** | **0.1044 (s)** | **0.2236 (s)** | **0.0337 (s)** | **0.0164 (s)** | **0.1466 (s)** |
>
> Based on the empirical results, we observe that the computational cost is half of the current SOTA method, CoTA, and our performance is better than that of CoTA based on Tables 1 and 4. The extra experiment shows that our method is of practical use.
>
> (Please note that, since the above method shares the same collected responses set, we did not include the time used for multi-round generation for comparisons.)
>
> > The study reports automatic correlations but omits user studies that would show whether the uncertainty scores actually improve human confidence.
>
> Thanks for the comment, we agree that a set of human-labeled data could further improve the method’s convincingness. However, due to the limited time, it is hard to conduct human experiments during the rebuttal period; we will consider including the human experiment in future work.
>
> > The authors state in the limitations that less instruction‑tuned LLMs could undermine the protocol, yet they provide no empirical evidence, but we need to see that. An appendix evaluating a lightweight model such as Llama3‑1B would clarify how sharply Topo‑UQ’s performance diverges from baselines (CoTA, Embed‑UQ, Entail‑UQ, NLI‑logit) under low‑capacity settings.
>
> Thanks for the suggestion, we have an experiment to summarize as below:
>
> Based on the literature, and by our experiment, the instruction-following ability for candidate models are in ranking:
> GPT4o > Llama 70b > DeepSeek (R1, 70b distilled version) >  Phi4 > Llama3-8b.
>
> |     Instruct-following ability      | GPT4o>  | Llama3 70b> | DeepSeek (R1, 70b distilled version)> | Phi4>  | Llama3-8b |
> |:-----------------------------------:|:-------:|:----------:|:-------------------------------------:|:------:|:---------:|
> | **Success Rate (GAM8K)**            | 0.9800  | 0.8655    | 0.4775                                | 0.1710 | 0.1287   |
> | **Success Rate (BoolQ)**            | 0.9426  | 0.8906    | 0.2797                                | 0.2606 | 0.2274   |
> | **Success Rate (GeoQA)**            | 0.8889  | 0.8424    | 0.3258                                | 0.1559 | 0.1061   |
>
> By focusing on the same group of models: Llama3 70b vs Llama3-8b, we find that, the lower success rate model (Llama3-8b) is aligned with the empirical results that: Llama 3-8b is worse than Llama3 70b in terms of the instruct-following abilities (also as in Figure 6). It is aligned that: Topo‑UQ’s on Llama 3-8b (-0.14, -0.13, -0.08) is worse than the same for Llama 3-70b (-0.43,-0.41, -0.28) across the metrics: PCC, SRC, KR  (↓) [results from GSM8K].

---

> > ### Author Response · Authors · 2025-06-03
> > **Continue. Reply to Reviewer GC6U**
> >
> > We conduct further experiments as suggested by the reviewer using the open-sourced model Llama3‑1B (meta-llama/Llama-3.2-1B) and compare it with the existing two larger models on the same dataset.
> >
> > |   Method   | Llama3-70B PCC | Llama3-70B SRC | Llama3-70B KR | Llama3-8b PCC | Llama3-8b SRC | Llama3-8b KR | Llama3-1b PCC | Llama3-1b SRC | Llama3-1b KR |
> > |:----------:|:--------------:|:--------------:|:-------------:|:-------------:|:-------------:|:------------:|:-------------:|:-------------:|:------------:|
> > |    CoTA    |     –0.13      |     –0.13      |     –0.09     |     –0.04     |     –0.03     |    –0.02     |     0.02      |     0.09      |    -0.02     |
> > |  Embed-UQ  |      0.15      |      0.14      |      0.10     |      0.23     |      0.23     |     0.15     |     0.05      |     0.01      |    0.02     |
> > | Entail-UQ  |      0.15      |      0.13      |      0.09     |     –0.08     |     –0.07     |    –0.05     |     -0.02      |     0.06      |    0.01     |
> > |  NLI-logit |      0.13      |      0.12      |      0.08     |      0.21     |      0.19     |     0.13     |     0.01      |     0.00      |    0.4     |
> > |    Ours    |     –0.43      |     –0.41      |     –0.28     |     –0.14     |     –0.13     |    –0.08     |     -0.03      |     0.01      |    -0.03     |
> >
> >
> > We find that the smaller (response generation) model with limited instruction-following performance results in difficulty in evaluating their performance by elicitation. Thus, we would recommend applying our method in those models with relatively strong instruct following abilities. In this example, we suspect that the Llama3-1b's limitation partially lies in the it’s maximum output token length (2048 tokens), while the Llama3-8b is 8192 tokens for each output.
> >
> >
> > Finally, we appreciate the reviewer's recognition of this work, and we'd also like to address any other potential confusions!

---

> > > ### Comment · Reviewer_GC6U · 2025-06-03
> > >
> > > Thank you for answering my questions and conducting the additional experiments. This confirms the quality of this paper and I am in favor of acceptance.

---

### Official Review · Reviewer_ymzu · 2025-05-12

**Rating:** 7
**Confidence:** 3
**Ethics Flag:** 1

**Summary:**

This paper introduces a new measuring method for qualifying uncertainties in LLM explanations. The proposed method takes into consideration the typological structures of LLM reasoning processes, attaining finer-grained analyses compared to previous methodologies. Experimentation is conducted on five major LLMs with three datasets, showing the advantage of the proposed method.

**Questions To Authors:**

Please discuss more details about the limitations of the reasoning typologies mentioned above.

After the Knowledge Point Reflection, how is it guaranteed that each of the sub-questions can be directly answered by the LLM? Or, isn't it a cause of a limitation to the proposed method?

In Definition 2, the nodeRaw and nodeResult should be defined in advance. How do you define the valid path in the reasoning? Are they the paths that connect the initial question and the correct answer?

**Reasons To Accept:**

The paper proposes a new framework for evaluating reasoning uncertainty in LLMs.
The proposed method considers finer-grained topological reasoning steps of LLMs.
It conducts detailed experimental evaluation of the proposed method comparing with existing baselines.

**Reasons To Reject:**

1. The proposed method mainly concerns the query and answer pairs that require multiple reasoning steps.

2. The method first asks an LLM to generate knowledge points, sub-questions, that comprise reasoning steps from the query to the answer, then asks an LLM to generate answers for the sub-questions. As the type of reasoning is limited, the type of uncertainties might be limited to this form of reasoning. For example, the Chain of Thought reasoning, which is a dynamic sequence (a new step takes into consideration the previous whole steps) may generate different forms of reasoning steps on the same queries.

---

> ### Author Response · Authors · 2025-06-02
> **Reply to Reviewer ymzu**
>
> **Thank you for the review of our work! We will answer the questions and weaknesses in the following**:
>
> > The proposed method mainly concerns the query and answer pairs that require multiple reasoning steps.
>
> Our method mainly focuses on the multi-step reasoning tasks, since it is critical for understanding the model's step-wise performance in decision making, however, the proposed Topo-UQ does not require the input to be multi-step reasoning.
>
> If provided with single-step answering, our method can still handle the problem by {nodeRaw-Edge-nodeResult} structure, where nodeRaw is the raw question, Edge could be None, and nodeResult is the direct response of LLM. It is worth noting that, in this scenario, the results are more like semantic variance, since the reasoning structure is monotonous.
>
>
> > The type of reasoning is limited, the type of uncertainties might be limited to this form of reasoning
>
> The way we construct knowledge points (sub-questions) and sub-answers is a superset that covers the chain-of-thought (CoT); if CoT structured is elicited, our method can still apply to the graph structures where edges do not have text (directly reasoning connected by: step1, step2 ...), in this case, Eq.(3) only works on node embedding and further GED procedure becomes the measuring the semantic variance e.g., semantic entropy.
>
> > After the Knowledge Point Reflection, how is it guaranteed that each of the sub-questions can be directly answered by the LLM?
>
> Thanks for the question. In the reasoning topology construction process, we have three modules working together: (1) Knowledge Point Reflection,  (2) Self-Answering Module, (3) Reasoning Topology Construction. As introduced in line 157, of ` (2) Self-Answering Module`, to guarantee each of the questions is answered by the LLM itself, the Self-Answering Module feed the sub-questions back to LLMs, and then provide sub-answers to the them; more details can be found in Appendix B (B.1 - B.3).
>
> > In Definition 2, the nodeRaw and nodeResult should be defined in advance. How do you define the valid path in the reasoning? Are they the paths that connect the initial question and the correct answer?
>
> We'd like to clarify on this question, the valid path does not emphasize the correctness, it is the path that connects the initial question to the final answer of the LLM. It means that the answer could be either correct or wrong.
>
>
> We appreciate the reviewer for recognizing the contribution of our work, and we are more than glad to engage in any further discussions!

---

> > ### Comment · Reviewer_ymzu · 2025-06-07
> > **Thank you for your clarifications**
> >
> > Thank you for answering my questions and concerns properly. I keep my evaluation in favor of acceptance.

---

### Official Review · Reviewer_oZN2 · 2025-05-13

**Rating:** 7
**Confidence:** 4
**Ethics Flag:** 1

**Summary:**

This paper introduces a novel framework (Topo-UQ) to quantify uncertainty in LLM explanations by explicitly modeling the reasoning process as a graph topology. The authors built on prior work on uncertainty quantification treating explanations as flat text. To improve on the flat structure, Topo-UQ captures both semantic and structural uncertainty using LLM-generated subquestions and subanswers to construct a directed reasoning graph. It then uses a modified graph edit distance metric (Reason-GED) to quantify variance across multiple reasoning paths and identify redundancy. The authors carry out experiments across three datasets (GSM8k, BoolQ, GeoQA) and five LLMs (GPT-4o-mini, LLaMA-3, DeepSeek-R1, Phi4): Topo-UQ shows stronger negative correlations between uncertainty and faithfulness compared to several baselines (Chain-of-Thought, Embedding-based, Entailment probability-based, logit-based).

**Questions To Authors:**

You use standard BERT embeddings to encode reasoning steps and subquestions. Could you elaborate on why this model was chosen over more recent alternatives, and how reasoning could be included into these semantics?
Does the method scale well when the number of reasoning steps (nodes) increases significantly, or does graph complexity become a bottleneck?
Could the redundancy metric be used not only for analysis but also for fine-tuning?

**Reasons To Accept:**

The analysis is well carried out and well evaluated across several models and baselines. Also the authors evaluate across several axes, including faithfulness, redundancy, as well as analyzing the reasoning patterns (embedding and clustering) to identify different reasoning types. The graph edit distance metric captures both of the semantics (embeddings) and graph differences.
The paper is well motivated, well written, and well presented.
While analyzing a graph structured output to me seems like a logical step in evaluations, to the best of my knowledge it is the first time it has been carried out on reasoning explanations.

**Reasons To Reject:**

Correctness vs. Uncertainty: The framework measures consistency/variability in reasoning, but does not directly assess or evaluate correctness.
The embeddings analysis relies solely on BERT, as there are more, and as I find more advanced embedding models, it would be interesting to compare those, E.g., different embedding models.
While this is an important work for interpretability research, my main concern, however, is that the approach is not evaluated using any downstream task performance. To see how applicable and generalizable the framework is, I find this to be important, also as a way to measure correctness vs. uncertainty.

---

> ### Author Response · Authors · 2025-06-03
> **Reply to Reviewer oZN2**
>
> **We appreciate the reviewer for the thoughtful feedback, and we’d like to address the questions as below**:
>
> > Q1: Why was the Bert embedding model chosen over more recent alternatives?
>
> 1. The BERT was selected as our embedding model because it is a widely adopted, general-purpose semantic encoder, which allows us to focus squarely on demonstrating our method’s ability to capture both semantic and topological uncertainty.
>
> 2. Besides, we conducted further experiments from Deepseek-R1 and used three different embedding models (Bert, RoBERTa, and DeBERTa) as advised by the reviewer, and we find our method's results are consistent and robust to various embedding models:
>
> **Dataset: BoolQ**
> |   Method  | DeepSeek-R1 PCC (Bert)↓ | DeepSeek-R1 SRC (Bert)↓ | DeepSeek-R1 KR (Bert)↓ | DeepSeek-R1 PCC (RoBERTa)↓ | DeepSeek-R1 SRC (RoBERTa)↓ | DeepSeek-R1 KR (RoBERTa)↓ | DeepSeek-R1 PCC (DeBERTa)↓ | DeepSeek-R1 SRC (DeBERTa)↓ | DeepSeek-R1 KR (DeBERTa)↓ |
> | :-------: | :---------------------: | :---------------------: | :--------------------: | :------------------------: | :------------------------: | :-----------------------: | :------------------------: | :------------------------: | :-----------------------: |
> |    CoTA   |          –0.22          |          –0.20          |          –0.13         |            –0.25           |            –0.21           |           –0.14           |            -0.22           |            -0.21           |           -0.14           |
> |  Embed-UQ |           0.56          |           0.56          |          0.39          |            0.06            |            0.07            |            0.05           |            0.15            |            0.15           |           0.10           |
> | Entail-UQ |           0.48          |           0.46          |          0.32          |            0.48            |            0.46            |            0.32           |            0.49            |            0.45           |           0.31           |
> | NLI-logit |           0.56          |           0.56          |          0.39          |            0.52            |            0.53            |            0.37           |            0.48            |            0.47           |           0.33           |
> |  **Ours** |        **–0.29**        |        **–0.26**        |        **–0.17**       |          **–0.29**         |          **–0.26**         |         **–0.17**         |          **-0.31**         |          **-0.28**         |         **-0.19**         |
>
> 3. Kindly note, as explained in Line 180, embedding quality may vary across domains, choosing the embedding model to best fit the domain needs can better improve real-world performance.
>
> > Q2: How could reasoning be included into these semantics?
>
> In our framework, after eliciting the reasoning graph, which contains sub-questions and sub-answers as texts, we apply the embedding function to each node and edge to capture the semantic meaning (in step 1 at Section 3.2), where each node and edge will be represented as embeddings. In general, we would say it is more like “semantics are included in the reasoning graphs”, not “reasoning is included in the semantics”.
>
> > Q3: Does the method scale well when the number of reasoning steps (nodes) increases significantly?
>
> It is a great question. If the number of graph nodes increases significantly, the classic GED we refer to will face a bottleneck of complexity. However, based on our experiment on three commonly adopted datasets: GSM8K, BoolQ, and GeoQA, the reasoning process takes at most 12 steps, and the current Topo-UQ handles the computation efficiently. Even though applying to the more difficult dataset, the reasoning steps might increase, however there is an active research direction on optimizing the GED efficiency and an alternative way to solve is by using an `Approximated GED` [1] [2], which could reduce the computing complexity.
>
> [1] Riesen K, Ferrer M, Fischer A, et al. Approximation of graph edit distance in quadratic time[C]//International Workshop on Graph-Based Representations in Pattern Recognition. Cham: Springer International Publishing, 2015: 3-12.
>
> [2] Chang L, Feng X, Yao K, et al. Accelerating graph similarity search via efficient GED computation[J]. IEEE Transactions on Knowledge and Data Engineering, 2022, 35(5): 4485-4498.
>
> > Q4: Could the redundancy metric be used not only for analysis but also for fine-tuning?
>
> Yes indeed, this is a promising direction that one could leverage the designed metric in this paper to construct a dataset containing the original reasoning topology and de-redundancy reasoning topology; in this way, with appropriate fine-tuning design, the language model could improve the reasoning efficiency by learning from two contrastive reasoning processes and leaning towards the efficient pattern.
>
>
> We are grateful for the positive feedback and the suggestions to improve our paper. We are happy to provide further explanations if needed!

---

> > ### Comment · Reviewer_oZN2 · 2025-06-10
> > **Response to Authors**
> >
> > Thank you very much for your detailed response! I’ll keep my score suggesting to accept the paper.

---

### Decision · Program_Chairs · 2025-07-08

**Decision:**

Accept

**Comment:**

This paper proposes a novel method that "parses" a CoT into a reasoning graph, which provides a more structured view of how these models perform search and reasoning. It would be a very useful tool for understanding and interpreting long CoTs from reasoning models like o1 and r1. It may also enable other methods that work on graph structure instead of flat text. The main downside is how well this method scale to larger problems. Reviewers unanimously recommended acceptance.